

# Characterisation of short-term extreme methane fluxes related to non-turbulent mixing above an Arctic permafrost ecosystem

Carsten Schaller[1,2,*], Fanny Kittler[2], Thomas Foken[1,3], and Mathias Göckede[2]

[1]University of Bayreuth, Department of Micrometeorology, 95440 Bayreuth, Germany
[2]Max-Planck-Institute for Biogeochemistry, 07745 Jena, Germany
[3]University of Bayreuth, Bayreuth Center of Ecology and Environmental Research (BayCEER), 95440 Bayreuth, Germany
[*]now: University of Münster, Institute of Landscape Ecology, Climatology Group, Heisenbergstr. 2, 48149 Münster, Germany

*Correspondence to:* Mathias Göckede (mathias.goeckede@bgc-jena.mpg.de)

**Abstract.** Methane ($CH_4$) emissions from biogenic sources, such as Arctic permafrost wetlands, are associated with large uncertainties because of the high variability of fluxes in both space and time. This variability poses a challenge to monitoring $CH_4$ fluxes with the eddy covariance (EC) technique, because this approach requires stationary signals from spatially homogeneous sources. Episodic outbursts of $CH_4$ emissions, i.e. outgassing in the form of bubbles from oversaturated groundwater or surfacewater, are particularly challenging to quantify. Such events typically last for only a few minutes, which is much shorter than the common averaging interval for eddy covariance (30 minutes). The steady state assumption is jeopardized, which potentially leads to a non-negligible bias in the $CH_4$ flux. We tested and evaluated a flux calculation method based on wavelet analysis, which, in contrast to regular EC data processing, does not require steady-state conditions and is allowed to obtain fluxes over averaging periods as short as 1 minute. We demonstrate that the occurrence of extreme $CH_4$ flux events over the summer season followed a seasonal course with a maximum in early August, which is strongly correlated with the maximum soil temperature. Statistics on meteorological conditions before, during, and after the detected events revealed that it is atmospheric mixing that triggered such events rather than $CH_4$ emission from the soil. By investigating individual events in more detail, we identified various mesoscale processes like gravity waves, low-level jets, weather fronts passing the site, and cold-air advection from a nearby mountain ridge as the dominating processes. Overall, our findings demonstrate that wavelet analysis is a powerful method for resolving highly variable flux events on the order of minutes. It is a reliable reference to evaluate the quality of EC fluxes under non-steady-state conditions.

## 1 Introduction

Methane ($CH_4$) is one of the most important greenhouse gases (Saunois et al., 2016b), but unexpected changes in atmospheric $CH_4$ budgets over the past decade emphasise that many aspects regarding the role of this gas in the global climate system remain unexplained to date (e.g. Saunois et al., 2016a; Nisbet et al., 2016; Schwietzke et al., 2016; Schaefer et al., 2016). Atmospheric $CH_4$ increased in concentration from 722 ppb in the year 1850, i.e. before industrialisation started, to 1810 ppb in the year 2012 (Hartmann et al., 2013; Saunois et al., 2016a). Current concentration levels are the highest reached in 800,000 years (Masson-Delmotte et al., 2013), and emissions and concentrations are likely to continue increasing, making $CH_4$ the





second most important greenhouse gas (after $CO_2$) that is strongly influenced by anthropogenic emissions (Ciais et al., 2013; Saunois et al., 2016b). In comparison to $CO_2$, $CH_4$ is characterised by a shorter atmospheric lifetime and a higher warming potential (34 times greater, referring to a period of 100 years and including feedbacks; Myhre et al., 2013). With management of $CH_4$ emissions being identified as a realistic pathway to mitigate climate change effects (Saunois et al., 2016a), quantitative

and qualitative insights into processes governing $CH_4$ sources and sinks need to be improved in order to better predict its future feedback with a changing climate.

The Arctic has been identified as a potential future hotspot for global $CH_4$ emissions (Zona et al., 2016), but the effective impact of rapid climate change on the mobilisation of the enormous carbon reservoir currently stored in northern high latitude permafrost soils remains unclear (e.g. Sweeney et al., 2016; Parazoo et al., 2016; Shakhova et al., 2013; Berchet et al., 2016).

Under warmer future conditions, increased thaw depths in Arctic permafrost soils as well as geomorphologic processes such as thermokarst lake formation are expected to mobilise carbon pools from deeper layers (Fisher et al., 2016), while at the same time the activity of methanogenic microorganisms may be promoted. Both factors would contribute to a potential increase in $CH_4$ emissions from permafrost wetlands (Tan and Zhuang, 2015). However, complex feedback mechanisms between climate change, hydrology, vegetation and microbial communities may partly counterbalance these increased emissions (Kwon et al.,

2017; Cooper et al., 2017). In order to improve the reliability of simulated Arctic $CH_4$ emissions under future climate scenarios, several process-based modeling frameworks for predicting $CH_4$ emissions have been improved in the last years (Kaiser et al., 2017; Raivonen et al., 2017), but the confidence in the results remains low, which can also be attributed to a lack of high quality observational data sets for $CH_4$ emissions from Arctic permafrost wetlands (Ciais et al., 2013).

The eddy covariance (EC) method allows for accurate and continuous flux measurements at the ecosystem scale, but strict

theoretic assumptions need to be fulfilled to ensure high-quality observations. Besides the requirement of steady-state conditions and a fully developed turbulent flow field (Foken and Wichura, 1996), the observation of $CH_4$ fluxes in high latitudes require some special considerations. These include technical challenges related to harsh climate conditions in remote areas of the high northern latitudes (Goodrich et al., 2016), and also problems related to atmospheric phenomena such as very stable stratification that inhibits turbulent exchange during polar winter. Methodological difficulties specific to $CH_4$ also play a role:

since net $CH_4$ emissions are not only dependent on the production conditions for $CH_4$ in the soil, but also on the transport processes from soil to atmosphere, they are characterised by higher temporal variability, compared to $CO_2$. $CH_4$ release through ebullition (Peltola et al., 2017; Hoffmann et al., 2017), i.e. episodic outgassing in the form of bubbles, typically occurs in events of only a few minutes in length, much shorter than the common averaging interval for EC (30 minutes). $CH_4$ ebullition events thus violate the steady state assumption for EC, with the potential to systematically bias flux calculations because of an

incorrect Reynolds decomposition. As a consequence, high emission events are likely to be discarded from the time series as very low quality data, or outliers, which has the potential to systematically underestimate long-term $CH_4$ budgets (Wik et al., 2013; Bastviken et al., 2011; Glaser et al., 2004).

To constrain potential systematic biases in EC data that are related to the afore-mentioned effects, a direct comparison with other observation techniques such as ecosystem chambers can be used. Experiments involving parallel observations with both

approaches have been conducted (e.g. McEwing et al., 2015; Emmerton et al., 2014; Sachs et al., 2010; Merbold et al., 2009;





Corradi et al., 2005). Chamber measurements are capable of resolving small-scale $CH_4$ emissions properly, but in most cases they cover only a small area on the order of up to a few $m^2$. Furthermore the installation of the chamber as well as its operation could introduce disturbances to the study area, which might lead to biased results. Upscaling approaches from the chamber to the EC footprint scale already exist (e.g. Zhang et al., 2012), but until now no method has been presented that aims to calculate

of $CH_4$ fluxes directly from high frequency EC measurements, including consideration of ebullition.

As a second approach to evaluate potential systematic biases in eddy-covariance $CH_4$ fluxes, a different calculation method can be applied to high frequency atmospheric observations that does not require the theoretic assumptions that limit the applicability of EC (Schaller et al., 2017b). Wavelet analyses provide this option (e.g. Collineau and Brunet, 1993a; Katul and Parlange, 1995), since they can be applied to calculate fluxes for time windows smaller than 10 to 30 min due to wavelet de-

composition in time and frequency domain without ignoring flux contributions in the low-frequency range. Moreover, wavelet transformation does not require steady-state conditions (Trevino and Andreas, 1996) but can also be applied on time series containing non-stationary power (e.g. Terradellas et al., 2001). As a drawback, the calculation of fluxes using wavelet transform requires considerably more computational resources even when a windowed approach is used.

The focus of the present study is on the interpretation of $CH_4$ emission events detected by a wavelet software package

(Schaller et al., 2017b, a), which has already successfully been applied to the non-steady state fluxes during a solar eclipse (Schulz et al., 2017). This approach, which builds on the raw data sampled by EC towers, allows us to resolve fluxes not only over 30 minute averaging periods, but also for an averaging interval of 1 minute. Such a higher temporal resolution facilitates detection of the exact time and duration of non-stationary $CH_4$ release events. The obtained results can be directly compared against EC fluxes, where a good agreement has been shown for times with well-developed turbulence conditions. We present

an analysis of whether peak $CH_4$ emission events at timescales on the order of minutes – triggered e.g. by ebullition – can be found in the results, what their basic characteristics are, and how these events may influence the computation of long-term $CH_4$ budgets. Finally the study aims to find meteorological triggers that could cause the observed events to occur.

## 2 Material and methods

### 2.1 Study site

Field work was conducted at an observation site within the floodplain of the Kolyma River (68.78° N, 161.33° E, 6 m above sea level) situated about 15km south of the town of Chersky in Northeast Siberia (Kittler et al., 2016; Kwon et al., 2017). The site is classified as wet tussock tundra underlain by continuous permafrost, with very flat topography. Averaged over the period 1960-2009, the mean annual temperature was -11°C, and the average annual precipitation amounts to 197mm (Göckede et al., 2017).

Two eddy-covariance towers were installed in summer 2013 about 600m apart, one of them (tower 1) focusing on an artificially drained section of the tundra site, the other (tower 2) serving as a control site to monitor undisturbed conditions. Both systems were equipped with the same instrumentation setup, including a heated sonic anemometer (uSonic-3 scientific, METEK GmbH) and a closed-path gas analyser (FGGA, Los Gatos Research Inc.), and feature about the same observation



height (tower 1: 4.9m a.s.l.; tower 2: 5.1m a.s.l.). Due to their proximity, both towers are also exposed to the same meteorological conditions. Inter- and Intra-annual variability of the exchange fluxes of $CO_2$ and $CH_4$, including an analysis of related environmental controls, are presented by Kittler et al. (2017b). For more details on the instrumentation setup, please refer to Kittler et al. (2016, 2017a).

## 2.2 Raw data processing and flux calculation

The raw data on the high-frequency fluctuations of wind and mixing ratios were collected using the software EDDYMEAS (Kolle and Rebmann, 2007) at a sampling rate of 20 Hz. Ancillary meteorological data were acquired at 1 Hz frequency through the LoggerNet software (Campbell Scientific Inc., Logan, Utah, USA) on a CR3000 Micrologger (Campbell Scientific). Both programs were running on-site on a personal computer, using the local time zone (Magadan time, UTC +12 h). The mean local solar noon is UTC + 13 h. Within the context of this study, datasets within the period 1st June to 15th September 2014 were analysed.

As a first approach to calculate turbulent $CH_4$ fluxes, we employed the eddy-covariance (EC) method using recent recommendations on correction methods and quality assurance measures (Aubinet et al., 2012). We used the software package TK3 (Mauder and Foken, 2015a, b) for this purpose, which includes all necessary corrections, data quality tests (Foken et al., 2012a), and a spike detection test using the Median Absolute Deviation (MAD / Hoaglin et al., 2000; Mauder et al., 2013). TK3 has been demonstrated to compare well with other available packages (Mauder et al., 2008; Fratini and Mauder, 2014). As the standard for the EC method, we derived turbulent fluxes with an averaging period of 30 minutes.

Because highly non-steady state conditions were expected for $CH_4$ fluxes at this observation site, which pose a serious violation of the basic assumptions linked to the eddy-covariance method (Foken and Wichura, 1996), we applied a wavelet-based calculation method as a second flux processing approach in addition to the standard eddy-covariance data processing. Schaller et al. (2017b) have developed a method for wavelet-based flux computation that offers the possibility of determining exact fluxes with a user-defined time resolution that can be as low as about 1 minute. Within the context of this study, we applied their calculation tool with a continuous wavelet transform using the Mexican hat wavelet ($WV_{Mh}$), which provides an excellent resolution in the time domain. Our results therefore allow an exact localisation of single events in time (Collineau and Brunet, 1993b). We additionally processed the data using the Morlet wavelet ($WV_M$), which provides an excellent resolution in frequency but a worse resolution in time domain, compared to Mexican hat (Domingues et al., 2005). In combination with the Mexican hat wavelet, this additional information can provide additional insight into turbulent flow characteristics, and therefore a better characterization of highly non-stationary datasets For steady state conditions, the wavelet and eddy-covariance methods have been shown to be in very good agreement. For more details on the implementation of the method directly refer to Schaller et al. (2017b).



## 2.3 Detection and classification of events

### 2.3.1 Detection of events

While spikes within the 20 Hz raw data were already identified in the MAD-Test (Mauder et al., 2013), in a first stage of the
wavelet-based event detection we conducted an additional MAD test on processed fluxes similar to Papale et al. (2006):

$$\langle d \rangle - \frac{q \cdot MAD}{0.6745} \le d_i \le \langle d \rangle + \frac{q \cdot MAD}{0.6745} \tag{1}$$

where

$$d_i = (x_i - x_{i-1}) - (x_{i+1} - x_i) \tag{2}$$

parameterises the difference of the current value $x_i$ to the previous and next value in time. $\langle d \rangle$ denotes the median of all those
double differenced values and

$$MAD = \langle |d_i - \langle d \rangle| \rangle \tag{3}$$

Due to its robustness the median absolute deviation is a very good measure of the variability of a time series and substantially
more resilient to outliers than the standard deviation (Hoaglin et al., 2000). The test was applied to Mexican hat wavelet flux
with a time step of $\Delta t = 30\,\text{min}$. If a value $d_i$ in the time series exceeded the given range in equation (1), it was detected as
an event. A threshold value of $q = 6$ was found to be suitable to reliably separate events from periods with a regular exchange
flux between surface and atmosphere.

The same MAD test calculations have also been applied to the flux with averaging interval $\Delta t = 1\,\text{min}$. The purpose of this
higher resolution analysis was first to precisely constrain the duration of an event down to the resolution of minutes, and second
to allow the detection of exact start and end times of events. We defined here a minimum duration of 2 minutes for an event,
since this way we could avoid labelling a sequence of high-frequency spikes, which sometimes pass the TK3 spike detection
threshold, as an event.

### 2.3.2 Classification of events

The approach described above only detects 1-minute-steps belonging to an event, but does not provide any knowledge about
typical structures of such contiguous single events. The term "structure" in this context refers to the specific sequence of
consecutive 1-minute flux values that together form the event: in a simple case, flux rates regularly increase until reaching a
plateau, then drop back to their starting values, with no events directly before or afterwards. More complex events appear as
clusters, i.e. during a prolonged period of time several shorter events occur close to each other. Since events with different
structure may also be triggered by different atmospheric conditions, we developed a basic classification to differentiate types
of events consisting of adjacent 1 minute steps.

Based on the single event-minutes identified by the MAD test, a manual search for characteristic, repeating patterns within
all half hour intervals that contained events resulted in the definition of three typical event structures. In this context, it was



found that the MAD test for a threshold value of 4 or 6 was not always able to resolve the whole event (blue plus-sign within grey shaded event duration in Fig. 1), thus in such cases actual starting and ending time of an event were corrected manually.

We labeled the first event type a single **peak event**. For this category, in the simplest case the flux increased monotonically up to one maximum event peak or a plateau with high flux rates, followed by a monotonic decrease back to base level. No other

events were detected within 30 min before or after the single peak event. As the example (Fig. 1, top panel) shows, such an ideal sequence cannot be expected in general, but in all cases a pattern of coherent single event-minutes showing the tapering to one peak or a few subsequent local maxima clearly suggested the classification of a "peak event". Peak events can occur as either negative or positive outliers from the baseline flux. If a positive peak was followed or preceded by a negative one or vice versa, both were combined into a single "peak event" as long as the magnitude of the second peak was lower than one quarter

that of the main peak.

We termed the second event class **down-up** events. Down-up events had the same basic properties as single peak events, but in contrast they consisted of one sharp negative and positive peak each, which were of similar magnitude (Fig. 1, center panel). If the order of the two peaks was reversed, the process was called an **up-down** event. Typically the two extremes within a down-up event were separated by several minutes (e.g. 4:58 and 5:01 in Fig. 1, middle), and such transition periods were

frequently not labelled as events by the MAD test because they did not exceed the threshold for event detection. In this case these event-minutes needed to be manually added to form a coherent down-up event.

The third class of events in our classification scheme was called **clusters**. In this category we collected all events that did not meet the criteria defined above for single peak events or down-up events, instead showeing a coherent pattern but not an unambiguous structure. This was generally the case for longer event periods that were potentially formed by the merging of

several consecutive shorter events (Fig. 1, bottom panel). However, in these cases a clear distinction of individual events was impossible due to the close succession of events over time, and the associated partial overlap. Accordingly, the identification of meteorological triggers for single events (see also Section 3.4) was also impeded, since more than one trigger may have been involved. We therefore handled the classification of events very conservatively, assigning single peak or up-down/down-up events only in very clear cases, while all remaining events were labeled clusters.

**2.3.3    Linking events to meteorological conditions**

For all events detected within the observation period, computed flux rates as well as prevalent meteorological conditions before, during, and after the event were collected in a database. These conditions were available as parameters in four different aggregation time steps: (1) $CH_4$ flux rates from both EC and wavelet processing as well as friction velocity ($u_*$) were used at 30 minute intervals. (2) Longwave radiation budget ($I$), air temperature ($T$), relative humidity ($R$) and air pressure ($p$) came

in 10 minute time steps. (3) 1-minute $CH_4$ flux rates were available from the high-resolution wavelet processing. Finally (4) wind speed ($U$), $CH_4$ mixing ratios ($c_{CH_4}$) and wind direction ($WD$) were taken from 20 Hz raw data. Averages for the period during the event were aggregated between start and end times of the detected event, while for the periods before and after the event mean values were derived for 10 minute intervals before the event start or after the event end, respectively. Regarding the





coarser resolution datasets (1) and (2), in each case the time step that overlapped most with the target timeframe before, during and after the event was chosen.

## 3  Results

### 3.1  Event statistics

Most statistics in this section are based on the number of minutes that were identified as part of an event. Using a flux averaging interval of $\Delta t = 1\,\mathrm{min}$, these minutes were defined as values failing the MAD test. For this analysis, the study period from 1[st] June to 15[th] September 2014 was split into seven blocks with a length of half a month each.

Our event detection algorithm identified 49 events for each site during the given observation period. 28 (tower 1) and 23 (tower 2) of these events were classified as clusters, while at both towers 6 events showed the typical shape of an up-down or
down-up event. Including interpolation between event minutes detected by the MAD test, the cluster events covered a combined period of 65 (tower 1) and 49 hours (tower 2), with a minimum duration of 49 and 31 minutes, and a maximum duration of 410 and 329 minutes. All clusters and up-down/down-up events occurred exclusively during nighttime.

The remaining 15 (tower 1) and 20 (tower 2) events were characterised as single peak events. Only 4 of these occurred during daytime (12.06. and 15.06.14), while all other events occurred at night. The duration of these peak events ranged between 2
and 43 minutes, while about half of them lasted between 9 and 21 minutes. All peak events occurred simultaneously with an event at the other tower, i.e. a corresponding counterpart event at the other tower was observed at about the same time. We will subsequently refer to simultaneous events (one from each tower) as a "pair" of events, while "event" still denotes one event from a single tower. For 13 event pairs, both events were classified as "peak events", while the majority of the remaining peak and up-down events were paired with cluster events at the other tower.

The absolute number of detected event-minutes differed strongly between the two towers. At tower 1, their cumulative duration exceeded that observed at tower 2 by a factor of 1.4 (first half of September) to 2.8 (first half of August). As one example, in the first half of August 462 minutes were identified by the MAD test as being part of an event at tower 1, surpassing just 165 event-minutes detected at tower 2 by a wide margin. Summed up for the period 1[st] June to 15[th] September, a total of 1078 event-minutes were detected for tower 1, more than doubling the cumulative sum at tower 2 (539 minutes). An explanation
for this difference can be found in the statistical characteristics of the two datasets: tower 2 had 2.3 % more extreme outliers (values that exceeded the interquartile range by a factor of 1.5) compared to tower 1. As the median absolute deviation is resilient regarding outliers, the MAD test serves as a robust estimator that is only marginally influenced by these outliers.

### 3.2  Event seasonality

For both towers, the relative distribution of events over the summer season showed similar patterns: the largest proportion of
all events was detected in the first half of August (37.9 % and 30.6 % at towers 1 and 2). Earlier in the growing season, we observed a gradual increase in event occurrence from only a few percent in the first half of June to 19.3 % (tower 1) and 16.5 %





(tower 2) in the second half of July. Following the maximum in early August, the appearance of events decreased rapidly to a range between 5.9 % and 15.4 % per half month in late August – September.

Seasonal courses in event frequency appear to be linked to trends in soil thermal conditions, as indicated by e.g. the simultaneous drop in both event-minutes and mean soil temperatures in late August. At the control site, the median half monthly soil

temperature at $-8\,\mathrm{cm}$ depth gradually increases from 3.6 °C in the second half of June to its maximum at 5.1 °C in the first half of August, followed by the afore-mentioned steep drop to 3.3 °C in the second half of August (details e.g. in Kittler et al., 2016). Both the general shape of the seasonal course as well as the timing of the peak agrees with the detected seasonality in event flux percentages.

### 3.3  Links between events and meteorological conditions

The observation that peak events were exclusively detected simultaneously with an event at the other tower suggests that events are typically not triggered by local changes in soil effluxes, but rather by mesoscale meteorological effects. Due to their precise temporal delimitation, the class of peak events allowed a clear characterisation of conditions for the periods before, during and after events. Accordingly, based on the study of peak events we were able to correlate event occurrence with short-term shifts in meteorological conditions that may be responsible for triggering the observed peak events. The following paragraphs list

statistics on the most relevant potential influence factors.

The air temperature ($T$) measured at the top of the towers showed a monotonically decreasing trend in at least 60 % of all peak events (21 of 35). This temperature drift usually started more than 10 minutes before the event, and persisted until at least 10 minutes after the event. Temperature change in time in this context ranged between $-0.04\,\mathrm{K\,min^{-1}}$ within an 18 minute interval and $-0.27\,\mathrm{K\,min^{-1}}$ within a 22 minute interval. The opposite case of increasing air temperatures during a peak event

was detected only once. For the relative humidity ($R$) at the top of the tower, in at least 29 % (10 of 35) of all peak event cases a monotonic increase was observed within the timespan of at least 10 minutes before and after the event. Increase rates for this subset of events are within the range $+0.67\,\mathrm{K\,min^{-1}}$ within 9 min to $+0.86\,\mathrm{\%\,min^{-1}}$ within 22 min. To give an example, during the peak event that started on July 13 at 10:39pm, and had a total length of 22 $\mathrm{min}$, the temperature dropped by 5.9 K in total, while the relative humidity increased by 19 %. No case was observed where the relative humidity decreased significantly

during an event.

The wind speed ($U$) increased in 83 % of all cases (29 of 35) during a peak event, in comparison to the last 10 minutes before the occurrence. In 48 % (14 of 29) of these situations, however, $U$ decreased again right after the event. The largest increase in wind speed was found to be 7.4 $\mathrm{ms^{-1}}$, while for the majority of cases the difference between the time before and during the event ranged from 0.2 to 2.1 $\mathrm{ms^{-1}}$. The vertical wind speed ($w$), which is a direct part of all flux calculation

methods, remained very close to the ideal value of zero in all these cases. Still, minor variations within a very narrow range of absolute values showed a very similar pattern, i.e. in 74 % (26 of 35) of the peak events a temporary increase was observed, followed by a decrease in 54 % of these cases (14 of 26). The friction velocity ($u_*$) increased at the beginning of 94 % (33 of 35) of all peak events, and decreased again right afterwards in 76 % (25 of 33) of these cases. For half of these events, only a





moderate increase in the friction velocity was observed ($< 0.1$ to $0.3$ ms$^{-1}$), while the full range of shifts lay between $< 0.01$ and $0.7$ ms$^{-1}$.

For the stability of atmospheric stratification ($zL^{-1}$, with $z$ as measurement height and $L$ as Obukhov length), no general pattern for the conditions before, during and after a peak event could be found. In 43 % of all events (15/35) there was no

change in stability over time while the event occurred. For 7 cases, the stability during the 30 minute interval where the event occurred shifted towards more unstable stratification, while for 8 cases a change in the opposite direction was observed. About 23 % (8 of 35) of all events occurred during unstable stratification, exceeding the average data fraction of unstable stratification during night time (13 % for tower 1, 18.5 % for tower 2). The stability before, during and after daytime events was always neutral.

Summarising, since the majority of events were detected during the night, it could be expected that a large number of cases would be subject to systematically falling temperatures, and associated increases in relative humidity. On the other hand, the high percentage of peak events that are characterised by an increase and subsequent decrease in wind speed and friction velocity indicates that turbulence intensity in the atmospheric surface layer is a major influence factor. With a higher-than-average fraction of cases with neutral atmospheric stability associated with peak events, it can be speculated that such stratification

conditions promote the impact of sporadic increases in mechanically generated turbulence that lead to the high CH$_4$ emissions.

### 3.4    Case study: Nighttime advection

To demonstrate the characteristics of a typical peak event, as well as the approach we used herein to analyse and interpret it, the following sub-sections provide a detailed description of a case study during the night from August 2-3, 2014. We chose this particular event because conditions are well documented through photographs taken by the observer, which strongly support

our theory about the underlying triggering mechanism as described later in this section.

#### 3.4.1    Meteorological conditions during event period

Within the given night, at both tower 1 (Fig. 2) and tower 2 (similar general patterns, data not shown) no signs of an upcoming event could be registered until 11:30pm. Starting at 11:00pm, a light breeze from the southeast with a maximum wind speed around $1.5$ ms$^{-1}$ gradually decreased to a calm. The mean CH$_4$ concentration in this half hour was 2102 and 2112 pbb at

towers 1 and 2, and the friction velocity as a proxy measure for aerodynamically generated turbulent motion was very low ($< 0.1$ ms$^{-1}$). At 11:31pm, both towers registered an increase in CH$_4$ concentrations, associated with a minor increase of the wind speed. A temporary shift in wind direction to the northwest was reversed back to the southeast after a few minutes.

Around 11:45pm, the wind speed continued increasing to about $1.5$ ms$^{-1}$, and a few minutes later the wind direction turned to the east/northeast. The onset of the event itself was detected at 11:55pm (tower 1, Fig. 3) and 11:59pm (tower 2, Fig. 4),

and this period of high fluxes lasted until 0:18am (tower 1) and 0:07am (tower 2). During the time interval 23:30 to 23:59 when the event started, the half hourly averaged friction velocity $u_*$ increased substantially, disrupting the previously existing decoupling of surface and higher atmosphere due to stable stratification. This increased turbulence intensity potentially vented CH$_4$ pools that had accumulated near the ground towards the EC systems at tower top. Shortly after the end of the event, the





wind direction at both towers changed from the east back to the southeast, i.e. the same direction as before the event. The $CH_4$ concentrations also decreased. Wind speeds, on the other hand, did not decrease, while the friction velocity decreased marginally.

### 3.4.2 Wavelet fluxes during event period

The mean Mexican hat $CH_4$ flux rate during the event was calculated as $181\,\mathrm{nmol\,m^{-2}\,s^{-1}}$ at tower 1 (tower 2: 392). This value is substantially higher than the $7\,\mathrm{nmol\,m^{-2}\,s^{-1}}$ observed in the 20 minute period before the event (tower 2: 26) as well as the $19\,\mathrm{nmol\,m^{-2}\,s^{-1}}$ in the 20 minute period after the event (tower 2: 88). The relatively high mean flux rate after the event at tower 2 is caused by a short period of higher fluxes up to 0:20am. In addition to the average flux rates, the standard deviation of fluxes at tower 1 ($118\,\mathrm{nmol\,m^{-2}\,s^{-1}}$) also significantly exceeded the values before ($53\,\mathrm{nmol\,m^{-2}\,s^{-1}}$) and after ($31\,\mathrm{nmol\,m^{-2}\,s^{-1}}$) the event (tower 2 showed similar overall behaviour).

The exact times when the flux peaks occurred coincided with the highest energy and most positive contribution to the wavelet flux, as indicated in the wavelet cross-scalograms of both towers (Figs. 3, 4). Sensitivity studies revealed that the choice of the upper wavelet scale integration limit $J$ (Eq. (13) in Schaller et al., 2017b) and thus the maximum wavelet period $\lambda_{max}$ significantly impacts the flux computation: an extension of the upper period integration limit to $\lambda_{max} = 184\,\mathrm{min}$ (Mexican hat) as well as $\lambda_{max} = 190\,\mathrm{min}$ (Morlet) showed a significant increase of both wavelet fluxes. Still, we did not find any indication that hinted at an influence of gravity waves during this particular case study. The use of the Morlet wavelet (Fig. 5, top) resulted in a widely expanded area of high flux contribution over the whole study time from 23:00 to 1:00, where the low frequency periods from 40 to 180 min contributed most to the total flux. In addition, the detailed cross-scalogram up to a period of 30 min (Fig. 3, 4) demonstrated that the lowest period of substantial contribution was around 10 min. In contrast to the reduced resolution of the Morlet wavelet in time domain, the Mexican hat cross-scalogram (Fig. 5, bottom) generated a sharp temporal transition between periods of high and low flux contributions, and this separation allowed us to precisely constrain the duration of the event. The latter finding verified that the observed contribution in the Morlet cross-scalogram indeed originated from the event itself, and was not carried over from adjacent periods.

For all three flux processing approaches compared herein, average $CH_4$ flux rates for the 30-minute interval that contained the peak event are summarised in Table 1 for both towers. These results indicate that for the chosen event period, both wavelet methods yielded systematically lower fluxes compared to the EC reference. In contrast to a previous study that focused on conditions of well-developed turbulence with high quality data eddy-covariance data (Schaller et al., 2017b), in this case deviations from EC fluxes based on the Morlet wavelet ($EC - WV_M$ tower 1: 87 $\mathrm{nmol\,m^{-2}\,s^{-1}}$; tower 2: 66 $\mathrm{nmol\,m^{-2}\,s^{-1}}$) were larger than those found for the Mexican Hat ($EC - WV_{Mh}$: 52; 34).

Differences in the performance of both wavelet approaches can be explained by the different characteristics of the wavelets: as shown in Fig. 5, the better resolution in the time domain of the Mexican hat wavelet led to higher absolute flux rates that were constricted to a narrower time window compared to the Morlet wavelet. This phenomenon can also be observed in the cross-scalogram up to $\lambda = 30$ min (Figs. 3, 4), where Morlet results showed the same flux over a wider timespan than those for the Mexican hat. Despite the limitation of the maximum period to $\lambda = 33$ min, the Mexican hat may nonetheless resolve



flux contributions above that limit, because its resolution in frequency domain is worse compared to the Morlet wavelet. Therefore, as demonstrated by Schaller et al. (2017b), for longer-term flux integration the fluxes based on the Morlet wavelet should provide the most accurate results, while for this specific 30-minute time window the Mexican hat results should be most trustworthy. Differences from the eddy-covariance fluxes strongly suggest that regular eddy-covariance data processing
yielded biased results (i.e a systematic overestimation of fluxes) caused by non-stationary conditions.

### 3.4.3  Cold-air advection from mountains

Around 11:45pm, the first signs of a developing ground fog were observed and also documented by photographs. Additional pictures were taken during the following minutes near tower 2 (Fig. 6, top), i.e. around the time the events began. All pictures demonstrate a ground fog moving in from the northeast, where the ridges of two nearby hills, Mount Rodynka (351 m above
sea level) and Mount Panteleicha (632 m a.s.l., Fig. 7), are located. The time at which this fog reached tower 1 coincided well with the onset of the events. Shortly after midnight, another photograph (0:11am) demonstrates that the fog had largely disappeared, well aligned with the sharp decrease in flux magnitude that indicates the end of the event.

The observed ground fog was also reflected in the meteorological data (Fig. 2). During the slow buildup of the ground fog in the period between 11:20 and 11:50pm, the temperature at 2 m above the ground decreased by 1.3 K, while the relative
humidity showed a small increase in the same timespan. Within the same period, the longwave net radiation, which is a good measure of the temperature difference between the sky and the ground, decreased to minimum values of 23 $Wm^{-2}$, which implies a low temperature difference between the surface and the clouds, indicating very low clouds or fog.

### 3.5  Event triggers

Our statistics on meteorological conditions before, during and after the detected peak events reveal a common pattern for all
event situations, regardless of the mechanism that actually triggered the event: during a period of weak turbulence, the surface was at least partially decoupled from the lower atmosphere where the flux sensors were positioned. $CH_4$ that was emitted from the soil during this period could not properly be mixed up to the sensor level, therefore likely forming a $CH_4$-rich layer of air near the ground. In all event cases, either a general change in atmospheric conditions or a short-term meteorological phenomenon broke up the decoupling between the layers. As a consequence, the $CH_4$ pool in near-surface air layers was
vented up to the eddy-covariance level, and therefore detected as a pronounced peak in the flux rate.

This sequence of conditions strongly suggests that atmospheric mixing, and not $CH_4$ emissions processes from the soil, is the dominating mechanism behind the flux peak events as detected by our algorithm. Since we did not observe a single case study where a strong flux peak was detected within a previously well-mixed situation, our findings indicate that ebullition events, which can e.g. be detected at smaller scales with soil chambers (e.g. Kwon et al., 2017), usually are too small as
individual emissions, or not coordinated enough spatially across the relatively large footprint area (approx. $4000\,m^2$ at neutral stratification) to be detected by an eddy-covariance system. Following the detailed description of the case study presented in the preceding section, in the subsections below we briefly discuss several typical meteorological situations that were also observed to trigger events.

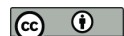



### 3.5.1 Cold air drainage

At the Chersky floodplain sites, about 50 % of all events occurred with wind directions from the E/NE, while only 3 % of all events fell into the S/SE (Table 2). These observations are in stark contrast to the local wind climatology, which lists just 16.2 % of cases in the E/NE sector, while the S/SE sector dominates with 37.9 % (values based on observations from tower 1, averaged for whole observation period). An explanation for this discrepancy can be found in the mesoscale wind field at this particular location, which may be prone to katabatic winds from the E/NE sector at night: typically, nighttime events from these sectors are characterised by decreases in the longwave net radiation $I$ to values around or below $20\ \mathrm{Wm^{-2}}$ exactly during or a short time after the event. This observation indicates that temperature differences between above and below the net pyrgeometer rapidly decreased, which could be a sign for low-level fog layers moving through.

### 3.5.2 Weather fronts

Weather fronts are typically associated with substantial shifts in e.g. air temperature, wind speed or wind direction. As an example, we observed such signs of a weather front passing the site on June 12, 2014, where the previously falling air pressure started increasing rapidly by 1 hPa per hour, combined with a wind speed increase from about 5 to $10\ \mathrm{ms^{-1}}$. With the stability of atmospheric stratification being neutral during this daytime event, it is unlikely that the mechanical turbulence associated with the frontal passage ejected $CH_4$ pools that had previously been accumulated close to the ground. Instead, it can be speculated that pressure fluctuations associated with the stronger turbulence washed out $CH_4$ from micropores within the top soil layers. However, particularly at night an accumulated $CH_4$ pool close to the surface should be the most likely source for a peak event, as e.g. observed during the night of June 13, 2014. Here the wind speed increased rapidly from about 1 to $4\ \mathrm{ms^{-1}}$, breaking up decoupled air layers between the surface and sensor level, and in the process venting the $CH_4$ that had previously been accumulated over time. This event was registered as rapidly shifting $CH_4$ mixing ratios at the tower top, which decreased within 10 min, while the wind speed continuously remained high.

### 3.5.3 Atmospheric gravity waves

For one pair of events occurring on July 12, 2014, conditions at both towers indicated low atmospheric turbulence intensity ($u_* < 0.3\,\mathrm{ms^{-1}}$), associated with a vertical temperature inversion and very low horizontal wind speeds. These conditions were interrupted at 3:10am, when the wind speeds first rapidly increased to $2.5\ \mathrm{ms^{-1}}$, only to drop to the previous low level ($0.5\ \mathrm{ms^{-1}}$) immediately afterwards. This step change was followed by both $CH_4$ concentration and vertical wind speed, where the former showed a sharp increase within seconds from around 2500 up to 5067 ppb (tower 1). For this situation, the Morlet crosswavelet spectrum showed a period around 5 to 10 minutes that contributed most to the observed flux. This information, together with the characteristics of the high frequency data, are indications that this particular event may have been triggered by an atmospheric gravity wave reaching the ground (Nappo, 2013; Serafimovich et al., 2017); however, lacking soundings of the vertical structure of the atmospheric boundary layer, this assumption remains speculative.



### 3.5.4 Low level jets

Low level jets appeared to be the triggering mechanism for two pairs of events with distinctive characteristics. In one example, on July 31, 2014, very low wind speeds ($0.5\,\mathrm{ms^{-1}}$) from NW to N resulted in a stably stratified lower atmosphere and a strong temperature inversion. In the period before the event occurred, the long-wave net-radiation decreased from about $30\,\mathrm{Wm^{-2}}$ to $< 15\,\mathrm{Wm^{-2}}$, which could indicate that low stratus clouds were moving in. The onset of the event itself was marked by a rapid increase of the wind speed and a shift in wind direction by at least $45°$ to S to SW, which led to a sharp rise in $CH_4$ concentration with maximum values around 4120 ppb (tower 1). The flux rate also substantially increased for 5 minutes. Within the next half hour, the wind speed gradually decreased, then the wind switched back to the direction before the event. Under nocturnal stable stratification with a typically shallow stable boundary layer, the observed sudden increase in wind speed in combination with a change in wind direction are indicators for a significant vertical wind shear associated with a low level jet, which was found to be connected with a significant increase in gas fluxes (Karipot et al., 2008; Foken et al., 2012b). But, as already mentioned for gravity waves, additional boundary layer measurements would be necessary to validate this assumption.

### 3.5.5 Onset of turbulent flow

The three remaining event pairs were detected under stable or neutral conditions, and characterised by a gradually increasing, non-fluctuating wind speed, but no change in flux rates just before the event occurred. One example from July 11 demonstrated that only when the increase in winds finally started to yield fluctuations in wind speed, the event occurred and the $CH_4$ concentration increased by about 500 ppb within 15 min. After the event peak was reached, the concentration decreased quickly, while the wind speed fluctuations did not change. These patterns indicate that, before the event, vertical decoupling of the shallow boundary layer resulted in a laminar wind flow at sensor height, which explains the dampened fluctuations in wind speed. With the shift from laminar to turbulent flow, the previously accumulated $CH_4$ near the ground could be transported up the sensor height, resulting in the observed flux peak. This observed change from laminar to turbulent flow is very similar to the conditions associated with a low level jet, but due to the missing shift in wind directions we decided to separate both triggering mechanisms herein.

## 4 Discussion

### 4.1 Advective contributions to flux events

The eddy-covariance method is based on the assumption that observations of turbulent fluctuations at a single point in space within the atmospheric surface layer can be used to obtain a representative flux rate from the ecosystem surrounding the flux tower. It is therefore of crucial importance for the interpretation of the impact of events for calculation of the local flux budgets whether the emitted $CH_4$ was produced locally and just temporarily pooled near the surface, or horizontally advected towards the measurement location. Advective transport would bias the local mass balance of $CH_4$ and any other atmospheric constituent to be monitored, therefore seriously undermining the theoretic assumptions that the eddy-covariance technique





relies on (Aubinet, 2008). If the fluxes detected by the instruments do not originate from the target area if advection is present, they should not be considered in the local flux budget. Accordingly, the detection of advection as a triggering mechanism behind an event deserves special attention, since inclusion of such data into the flux budget would lead to a systematic overestimation of fluxes from the local ecosystem.

To differentiate between events with and without advective flux contributions, the extension of the wavelet integration period provides essential information. For all methods compared herein, peak events are characterised by an intensive high-frequency turbulent component within an integration interval of up to 30 minutes, which explains the increase of the flux. In addition to this, events that were influenced by advection also showed significant flux contributions from longer integration periods. This finding indicates that the elevated flux rates were not exclusively driven by turbulence and the venting of local $CH_4$ pools near
the ground, but also contained contributions from mesoscale motions spanning periods of minutes to hours.

     The correlation in temporal trends of turbulence intensity and $CH_4$ mixing ratios after the event can also be taken as an indicator for the source of the $CH_4$. If the excess $CH_4$ that created a peak flux during a detected event was coming from a limited source, i.e. local emissions that had been pooled in air layers close to the ground, the increased $CH_4$ concentrations usually dropped to lower levels after only a few minutes. In this case, elevated flux rates also lasted for only a few minutes,
while the increased turbulent mixing that initiated an event often persisted for a long time thereafter. In contrast, if the triggering mechanism had been advective transport, both $CH_4$ concentrations and turbulence intensity should remain high for an extended period of time. Here, the reservoir that feeds the peak $CH_4$ fluxes is substantially larger, since it is originating from a different region and is transported to the tower by katabatic winds. However, the differentiation is not as clean as that based on the wavelet integration periods, since the maximum amount of $CH_4$ that can be vented from a local source close to the surface
in the absence of advective contributions depends on many factors. Most importantly, the time since the last event took place influences how much $CH_4$ can have re-accumulated, but the current $CH_4$ emission rate from the ground and the intensity of the vertical mixing with the onset of the event also play a role in how long it will take until a local source will be depleted. To summarise, based on the length of an event alone a clean distinction between events with and without advective flux contributions cannot be performed, but for events with a duration of 15 minutes or more, advection is likely to be present.

**4.1.1  Implications for designing an optimum observation strategy**

Statistics for the Chersky site show that, on average for the observation period in summer 2014, an event occurred about every other day (0.46 events per day). With the longer cluster events lasting for up to several hours, the average time covered by an event per day is 36.4 minutes at tower 1, and 27.5 minutes at tower 2. Assuming that such events, at best, lead to lower quality rating of the eddy-covariance fluxes, and in the worst case constitute systematic biases to flux budgets determined through
the eddy-covariance technique, their net impact on longer-term flux budgets may be substantial. Our results demonstrate two major pathways through which events can systematically disturb the flux budget determined through the conventional eddy-covariance approach:

     In the absence of advection, an event such as e.g. a peak event that produces a short but intense outburst of $CH_4$ with a duration of (significantly) less than the common integration interval for eddy-covariance (30 minutes) constitutes a substantial





violation of the steady-state assumption. As a consequence, the Reynolds decomposition that separates the high-frequency signal into a mean and turbulent component may produce incorrect positive and negative fluctuations of both vertical wind and trace gas concentrations. Depending on the nature of the event, the observation may in part be discarded as a spike, or the entire 30-minute interval may be flagged as very low quality data and in turn be sorted out during data analysis, to be replaced

by gap-filled values. In both cases, the high emission event would disappear from the long-term $CH_4$ flux budget, effectively leading to a systematic underestimation of net emissions. As a second potential scenario, the incorrect Reynolds decomposition may lead to both positive and negative flux biases, again dependent on the nature of the event, while a medium quality flag will lead to the inclusion of this flux into long-term budget computation. In summary, the presence of events will introduce additional uncertainty into long-term flux observations, and in the case of $CH_4$ is likely to lead to a systematic underestimation

of flux budgets since peak events are likely to be sorted out by the processing software.

As a second major pathway to disturb eddy-covariance flux budgets, events hold the potential to bias the local mass balance through advective flux contributions. Our statistics demonstrate that cold air drainage is the responsible trigger for about half of the peak events detected by our algorithm at the Chersky observation site. Wind statistics and regional topography structure support the assumption that these events are associated with horizontal advection of $CH_4$ that contributes a significant portion

of the excess flux. Based on overall event statistics, this means that the site experiences on average about 2-3 events per month with potential advective flux contributions during the growing season. For several reasons, the potential bias of this effect on the eddy-covariance flux budget cannot be quantified yet. First, the total flux during an event triggered by cold air drainage will be a composition of local $CH_4$ emissions pooled near the surface and advected $CH_4$. Second, a portion of the affected events will be sorted out by the eddy-covariance quality flagging procedure, and (in this case rightfully) removed from the long-term

budget computation. Therefore, as for the violation of steady-state conditions, advective events need to be considered as a potential cause for systematic biases, in this case overestimation, of eddy-covariance flux budgets.

To facilitate a differentiation between these pathways, it would be important to validate these mesoscale triggering mechanisms in future field experiments. Influences by low level jets or gravity waves could be verified by additional measurements of the atmospheric boundary layer, e.g. using a well established technique like SODAR/RASS (SOnic Detection And Ranging

/ Radio Acoustic Sounding System). The conceptual model of katabatic winds from the hill ridge located north / north-east of the study site could be investigated by installing additional nocturnal temperature measurements at heights of 20 to 50 cm in the hills and optionally also between the site and the hills. In order to visualise the events and to achieve a better understanding of how the accumulated $CH_4$ is mixed up to the sensor during an event, it could be helpful to use the high-resolution fibre-optic temperature sensing approach, which was newly developed by Thomas et al. (2012) and has already been established for

studies on cold air layers in the nocturnal stable boundary layer (Zeeman et al., 2015).

### 4.1.2   Role of cluster events

The potential role of events classified as "clusters" (coherent pattern, but no uniform shape) on potential systematic biases in flux budgets was excluded from this study. Clustered events, which made up the vast majority of event minutes detected by our algorithm, hold the potential to yield very different results between eddy-covariance and wavelet methods; however, a uniform




classification of e.g. environmental conditions and flux patterns was not possible here, therefore a detailed investigation needs to be postponed to a follow-up study that will be exclusively dedicated to this phenomenon. It is very likely that these clusters were a result of recurring events, and complex recirculation of air masses enriched in trace gases.

## 5 Conclusions

We showed that wavelet analysis can serve as a suitable method to resolve events in the order of minutes, which typically occurred at night and were not caused by ebullition or other local processes in the soil, but by different mesoscale meteorological phenomena. EC failed to resolve the events correctly.

In detail, this study demonstrates that events, which represent a violation of the basic assumption for the application of the eddy-covariance technique, are a regularly occurring phenomenon at the observation site Chersky in Northeastern Siberia. The
exact localisation of these events in time as well as their duration and magnitude was made possible using wavelet analysis.

All events evaluated in this study started with a similar general setting: $CH_4$, as emitted from the soil, accumulated near the ground because the surface layer was decoupled from the overlaying air during time periods of low turbulence. The breakup of these conditions was triggered by different mechanisms on the mesoscale. These mechanisms included the passage of fronts, atmospheric gravity waves, low level jets, and katabatic winds. All events were characterised by sudden peaks in $CH_4$ mixing
ratios, often connected with increased horizontal wind speeds. This led to turbulent mixing and thus to short-term events with increased $CH_4$ fluxes. We can rule out in virtually all cases that the observed peaks were the result of sudden, simultaneous $CH_4$ releases from the soil.

We found a strong positive correlation of short-term extreme $CH_4$ flux events during the season with high soil temperatures and high median $CH_4$ rates. This conjunction was likely formed by an increased $CH_4$ production during times of high soil
temperatures, which facilitated the accumulation of substantial $CH_4$ pools when the surface layer was decoupled from the air above. Further, we found that events that were triggered by katabatic winds, advected further $CH_4$ to the site, which must have been emitted at a remote place within the flow path of the advection. As half of all events within our dataset were linked to advection, the peaks therefore do not necessarily represent the characteristics of the local $CH_4$ production. This leads us to conclude that the respective flux events do not necessarily reflect the conditions at the site or within the EC flux footprint.

The portability of these results to other flux observation sites, within the Arctic and beyond, depends largely on prevalent local and regional atmospheric transport and mixing conditions. Particularly at sites where low winds at nighttime frequently enable an efficient decoupling of the surface layer, it is likely that similar phenomena may occur. The net impact of such events on the long-term $CH_4$ budget still needs to be quantified, particularly since a large fraction of events were present in the form of clusters that proved difficult to classify and analyse. Such an analysis will be subject of a follow-up study that is currently
in progress.

*Acknowledgements.* This work has been supported by the European Commission (PAGE21 project, FP7-ENV-2011, Grant Agreement No.
282700, and PerCCOM project, FP7-PEOPLE-2012-CIG, Grant Agreement No. PCIG12-GA-2012-333796), the German Ministry of Edu-



cation and Research (CarboPerm-Project, BMBF Grant No. 03G0836G), and the AXA Research Fund (PDOC_2012_W2 campaign, ARF fellowship M. Göckede). Furthermore the German Academic Exchange Service (DAAD) provided financial support for the travel expenses. Additionally we thank Andrew Durso for text editing.



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



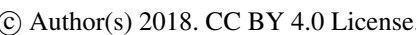



**Figure 1.** Examples for peak events (top), down-up events (center) and clustered events (bottom) identified using the Mexican hat wavelet flux. Data points marked with a yellow vertical line were detected as event-minute using the MAD test with threshold $q = 6$, while all other non-event data was marked with blue plus signs. The manually detected event length is shaded in grey colour.







**Figure 2.** Meteorological conditions observed at tower 1 during the case study event of August 02./03., 2014. Wind velocity $U$, vertical wind speed $w$ and $CH_4$ mixing ratio $c$ as well as wind direction $\phi$ are shown in a time resolution of 20 Hz. The friction velocity $u*$ was averaged to 30 min, while all other data were averaged to 10 min: relative humidity $R$ and air temperature $T$ (both in each 2.0 and 4.5 m above ground) as well as the longwave radiation balance $I\downarrow + I\uparrow$, the shortwave downwelling radiation $K\downarrow$ and air pressure $p$. The bottom panel shows a legend for $\phi$.





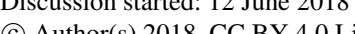

**Figure 3.** Wavelet cross-scalograms and flux rates computed for tower 1 during the case study event on August 02./03., 2014. The colours in the wavelet cross-scalograms between $w$ and $c$ denote the flux intensity, with warm colours indicating the highest flux contributions. Gray coloured intervals at $ITC$ and $RN_{Cov}$ refer to each best turbulent ($ITC < 30\%$) and steady-state ($RN_{Cov} < 30\%$) conditions according to Foken et al. (2004). The quality classes $1 - 9$ for EC refer to the overall flux flagging system after Foken et al. (2004).

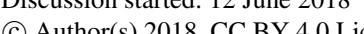



**Figure 4.** Wavelet cross-scalograms and flux rates computed for tower 2 during the case study event on August 02./03., 2014. The colours in the wavelet cross-scalograms between $w$ and $c$ denote the flux intensity, with warm colours indicating the highest flux contributions. Gray coloured intervals at $ITC$ and $RN_{Cov}$ refer to each best turbulent ($ITC < 30\%$) and steady-state ($RN_{Cov} < 30\%$) conditions according to Foken et al. (2004). The quality classes $1 - 9$ for EC refer to the overall flux flagging system after Foken et al. (2004).



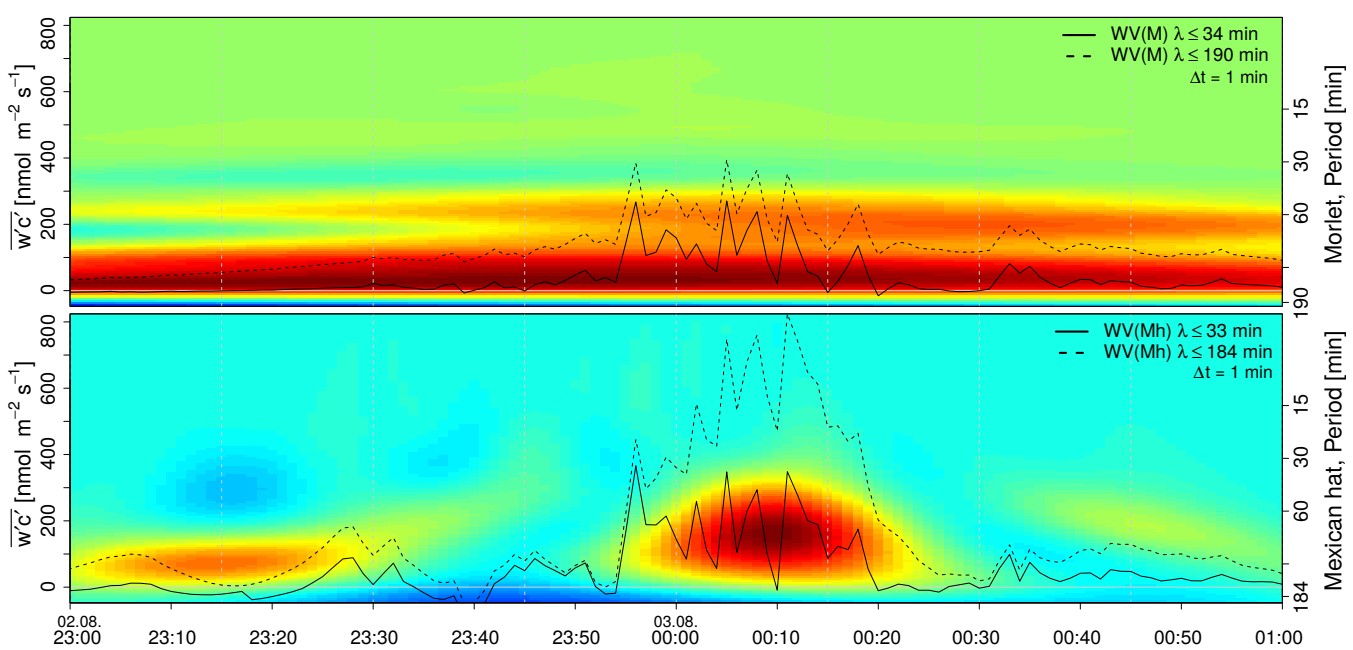

**Figure 5.** Wavelet cross-scalogram for the period 02.08.14, 11:00pm − 03.08.14, 1:00am at tower 1 for Morlet (top) and Mexican hat wavelets (bottom). The right axis numbers the period, while plotted lines refer to the left axis. Solid lines show the flux for an integration over all periods from $\lambda = 2 \cdot \delta t$ to $\lambda = 33\,\text{min}$ (Mh / Mexican hat) and $\lambda = 34\,\text{min}$ (M / Morlet), while the dashed line gives the flux up to $\lambda = 184\,\text{min}$ (Mexican hat) and $\lambda = 190\,\text{min}$ (Morlet). The colours in the wavelet cross-scalograms between $w$ and $c$ denote the flux intensity, where warm colours indicate the highest flux contributions.



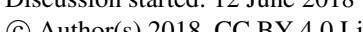

**Figure 6.** Photos of the study site directly at tower 1 (top and bottom left) and on the boardwalk between the power station at Ambolyka river and tower 2, taken between 2 Aug 2014, 11:57pm and 3 Aug 2014, 0:11am.



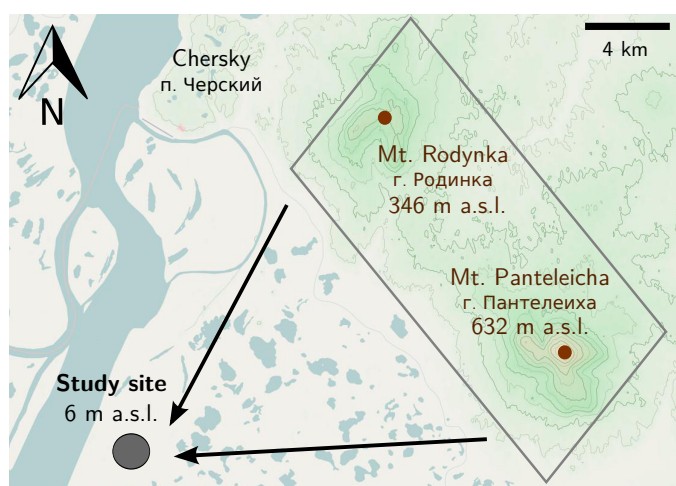

**Figure 7.** Flow path span of potential cold air drains from the ridge between Mount Rodynka and Panteleicha through the flat floodplains of Kolyma river to the study site (Map modified from www.openstreetmap.org (2015-06-24), copyright by OpenStreetMap contributors under Creative Commons License CC-BY-SA).



**Table 1.** Mean flux rates during the 30-minute period that hosted the peak event discussed in the case study, as detected by three different flux processing approaches. All flux values are given in $\mathrm{nmol\,m^{-2}\,s^{-1}}$.

| Approach | Tower 1 | Tower 2 |
|---|---|---|
| Eddy-covariance | 161 | 213 |
| Mexican Hat wavelet | 109 | 179 |
| Morlet wavelet | 74 | 147 |

**Table 2.** Night time frequency ($21 - 9$ o'clock) of the wind directions over the whole measuring period for both towers in percent. The last row gives the frequency of wind directions observed for nighttime peak events. Percentages greater than 20 % are underlined.

| Wind sector | N | NE | E | SE | S | SW | W | NW |
|---|---|---|---|---|---|---|---|---|
| Tower 1 | 25.3 | 8.1 | 8.1 | 29.2 | 8.7 | 3.6 | 4.5 | 12.4 |
| Tower 2 | 25.2 | 8.3 | 7.2 | 28.2 | 8.7 | 4.1 | 4.8 | 13.6 |
| During events | 13.3 | 20.0 | 30.0 | 3.3 | 0.0 | 10.0 | 6.7 | 16.7 |