# Peer review of "Characterisation of short-term extreme methane fluxes related to non-turbulent mixing above an Arctic permafrost ecosystem"

_Atmospheric Chemistry and Physics, 2018_

## Referee Comment (RC1) · N. Pirk (Referee) · 28 Jun 2018

The manuscript presents land-atmosphere methane fluxes from a permafrost-underlain wetland in NE Siberia, with a special focus on short-term fluctuations caused by non-turbulent conditions. The analysis uses wavelet transforms to calculate fluxes with higher time resolution than conventional eddy covariance calculations, also in non-turbulent conditions when the ground surface is (or has been) decoupled from the sensor level. Based on the wavelet flux time series, high-flux events are identified and classified according to their temporal structure. Some specific events are described in detail to distinguish the active mesoscale processes. The work should be interesting

for many in the eddy covariance community and is relevant for the scope of ACP. The language and figures are of good quality. I therefore recommend the publication of this manuscript after minor revision considering my comments below.

1. Generally speaking, it is important to have studies that present alternatives to the conventional eddy covariance flux calculations. After reading this manuscript, however, I am left with the impression that wavelet flux estimations give arbitrary flux results. You used two different wavelets, the Mexican Hat and the Morlet, and get flux differences by about 30% on average (cf. Table 1). Other wavelets might have given even larger differences. I think this makes it difficult for the wider EC community to apply the wavelet analysis approach. While I understand that you cannot resolve this arbitrariness in your manuscript, I think you should be clearer about the potential and the shortcomings of wavelet flux calculations.

2. You imply on several occasions that the high-flux events you identified are related to methane emissions from the ecosystem. For example, when you relate event occurrences to soil temperatures in the abstract (page 1, lines 9ff, "We demonstrate..."). Also many other parts of your manuscript are written as if this study is about bio-physical processes, and not just about a different flux calculation method. At the same time, I think you are aware that your events are probably not ecosystem emissions, but merely a venting of previously accumulated methane. The fluxes you present, e.g. in Figure 1, indicate fast shifts between methane emission and uptake, which are unlikely to have anything to do with the ecosystem dynamics. There is a tendency throughout your manuscript to smear out the distinction between the ecosystem methane exchange and the flux you calculate at sensor level.

3. On a related note, you mention that conventional EC processing would give biased budgets due to events of non-turbulent mixing, even if filtering and gap-filling is applied. However, you don't perform the comparison to a commonly-used filtering and gap-filling routine to make such a statement. If you want to say anything about such a possible bias, your analysis needs to show this.

4. You attribute parts of the high-flux events to methane that entered your footprint by horizontal advection. But since you have no direct measurements of horizontal methane advection, this attribution remains speculative in this study and should be phrased accordingly.

5. I think your dataset of methane EC measurements from NE Siberia is impressive and extremely valuable, but in this manuscript you don't use this potential very much: you say and conclude little about this ecosystem or the methane dynamics of it. I understand that you want to focus on wavelet analysis and short-term events, but you could probably have done this for much easier field studies (something like CO2 exchange above a central European farmland). You don't even mention your field site location in the abstract. I think there is room for improvement to integrate and connect your findings to methane flux studies from permafrost wetlands.

Specific comments:

6. Page 1, line 5 You mention that ebullition events last for only a few minutes, but I think the timescale of ebullition depends on the spatial scale. On a small spatial scale, maybe comprising a single bubble only, an ebullition event would probably only take seconds to be mixed into the ambient atmosphere.

7. Page 1, lines 12ff. "By investigating..." This sentence is unclear. You say you identified mesoscale processes as the dominating processes. But for what? You mean as the trigger for high-flux events? But then this is quite a stretch given your rather descriptive analysis of the mesoscale conditions.

8. Page 1, lines 15f. "It is a reliable..." Please elaborate and clarify this statement. How exactly can I evaluate the flux quality using wavelets? And did you show that this works reliably?

9. Page 4, line 22 What do you mean by "exact" fluxes?

10. Page 4, line 22 You mention the 1-min resolution of the flux results. But what is

the limit for the time resolution of wavelet flux calculations? Why can you not resolve 1 sec, for example?

11. Page 4, line 25 Is the Morlet wavelet you used a real or complex function? I'm asking because I think in the PyWavelets Python package, the "Morlet" wavelet is real-valued, which might be unexpected. At the same time, a real-valued wavelet might have the advantage that you can show the flux direction (uptake/release) in your cross-scalograms (cf. Figure 3).

12. Page 6, line 16 Here you mention that some event-minutes needed to be manually added. Later, in the last sentence of section 3.1, you describe the MAD test as a robust estimator. I wouldn't expect a robust test to need manual intervention.

13. Page 7, line 24 You explain the large difference between the two towers by the percentage of outliers. But what is the explanation for this difference in outliers? Could the real explanation be that tower 1's footprint was artificially drained?

14. Page 7, line 28 Here you describe the event seasonality, and I think it would be nice to see a plot of the event-percentages over time. You can show the three classes of events as separate lines, and the two towers as separate subplots. Maybe you can even add another subplot with the friction velocity.

15. Page 8, line 16 How did you quantify or identify a trend?

16. Page 9, line 34 I think it would be worthwhile to check if there is a relation between the length of the calm period preceding an event and the event's total emission. This test could give you a needed insight to separate local emissions from horizontal advection.

17. Page 10, lines 14f If an extension of the upper period limit changes the flux so much, does this mean there is no co-spectral gap? How does this problem look like during well-mixed, stationary conditions? Have you looked at the ordinary co-spectrum and ogive?

18. Page 11, lines 4f Wouldn't the regular EC processing filter out and gap-fill this period? If so, it doesn't seem right to say "regular EC data processing yielded biased results". And have you checked that the momentum flux is downwards for all these events you discuss here?

19. Page 11, lines 19ff This whole paragraph hits the nail on the head. You should focus on this finding in your abstract, instead of ebullition, which you probably didn't observe.

20. Page 14, line 20 Isn't it more the time since decoupling that determines how much methane can have accumulated, rather than the time since the last event?

21. Page 15, lines 5f Methane budgets with the ecosystem as a reference should not include such high-flux events, because the ecosystem did not emit these large amounts of methane in this period. So I fail to see that filtering and gap-fill these periods would lead to a systematic underestimation of net emission, as you state here.

22. Page 16, line 1 Why was this classification not possible here?

23. Page 16, line 7 I'm not sure EC really "failed to resolve the events correctly". It is not designed to resolve them in the first place.

24. Page 16, line 16 How did you rule out sudden sources from the soil?

25. Page 24, Figure 2 Your w-measurements seem to have a mean value of about 0.1 m/s, so this is data before the tilt correction? But your wavelet cross-scalograms use w after the tilt correction, right?

26. Page 25, Figure 3 The cross-scalograms don't seem to show a co-spectral peak, or an intensity decrease at the lowest and highest frequencies. Is this expected? Are these coefficients pre-multiplied by the frequency? Maybe a legend would help to read these plots. And did you define ITC and RNcov anywhere?

Technical corrections:

Page 4, line 28 Missing full stop.

Page 8, line 22 You probably mean +0.67 % min-1

Page 10, line 5 Please add units to the fluxes given in parentheses

Best wishes, Norbert Pirk

---

## Referee Comment (RC2) · Anonymous Referee #2 · 26 Jul 2018

Review of "Characterisation of short-term extreme methane fluxes related to non-turbulent mixing above an Arctic permafrost ecosystem" by Schaller et al. acp-2018-277

This manuscript presents an analysis of methane ($CH_4$) eddy covariance (EC) data measured above a wetland in NE Siberia. The manuscript focuses on $CH_4$ fluxes during night time in non-turbulent and low-mixing conditions when the EC measurement level is decoupled from the surface. Wavelet methods developed in a companion paper are used to estimate fluxes with 1 min time resolution over one summer and this high frequency flux time series is used to identify and classify high $CH_4$ flux events during

the analyzed period. These events are then speculated to be linked with atmospheric mesoscale circulation taking place in these nocturnal low-mixing conditions.

However, large part of the abstract, introduction and some other sections are discussing ebullition and other non-related topics, whereas results and conclusions are all about nocturnal low-mixing conditions. The authors should modify the beginning of the manuscript so that it matches with the end, so that the text forms one coherent entity. There are also other shortcomings in the text and description of data processing. Please see below.

As it stands the manuscript is interesting and shows promise but requires major revisions (see below) before publication. Once revised, it should be of interest also for the wider community working with micrometeorological flux measurements and hence the study is within the scope of ACP. Besides the shortcomings mentioned above, the presentation quality is good, although some figures need adjustment. I recommend the publication of this manuscript after major revision based on the comments below.

GENERAL COMMENTS

1) Please modify the abstract and introduction so that they match with the results. In my opinion these sections should be largely rewritten since now they are quite disconnected from the rest of the manuscript. The results are about gas fluxes under nocturnal low-mixing conditions and the abstract and introduction should be written about this topic, not about arctic wetland CH4 emission dynamics. As you know, these problems related to low-mixing conditions are universal, not only related to arctic wetlands.

2) The wavelet method is presented in the manuscript as more accurate than EC and reliable reference for the EC fluxes. This is a strong statement, which should be supported by convincing evidence. In order to make that kind of statement you should show that when the data is processed using standard EC methodologies spurious data points are left in the flux time series, yet with the wavelet methods these problematic periods are handled better. I suspect that most of the low-mixing conditions would have

been filtered out by quality screening and friction velocity filtering the data. Hence CH4 budgets derived using standard EC processing are likely not affected by these spurious fluxes during low-mixing. Please show a comparison of fluxes (e.g. monthly CH4 budgets) derived with standard EC processing (including quality screening and friction velocity filtering) and fluxes calculated with your wavelet methods to support your statement. Alternatively, you should phrase the text differently so that it is clear for the reader that it is not possible to say that the wavelet method is more accurate or reliable than standard EC.

3) In many occasions the observed high flux events are linked with mesoscale motions (gravity waves, low-level jets etc.), yet the connections are quite speculative. This is understandable since these mesoscale motions are difficult to quantify with just one flux tower and the authors also clearly state this in the discussion section of the manuscript. However in contrast to the discussion, the abstract and the conclusions are written in such way that the connections are obvious based on the data. Please rephrase the text so that it is clear for the reader that the role of mesoscale motions is quite speculative and additional instrumentation would be needed for a proper identification of these flow patterns.

4) More details about data processing are needed. Did you do coordinate rotation and how did you do it? The regular 2D-coordinate rotation (align u with mean wind and nullify mean w for each 30 min period) does not necessarily work well during low-turbulence since mean w is not necessarily zero when mesoscale motions are at play. Hence I hope that you used planar fitting (Wilczak et al., 2001) and defined the plane used in the coordinate rotation using high quality data. On the other hand if you did not do any coordinate rotation (like in Schaller et al., 2017) then the fluxes might be seriously compromised, since sonic anemometers are always slightly tilted no matter how carefully they are aligned with the surface below. Also did you correct for the time lag between gas analyzer data and sonic data? Lag time determination is always difficult for periods with low and intermittent turbulence. Therefore, please

provide additional details about EC processing. Related to the wavelet analysis, how did you take into account the time series edges and their effect on the results? Did you zero-pad the data and then estimate the cone of influence (Torrence and Compo, 1998)? This is important especially for the low frequencies. Please add details, since they are missing also from the companion paper (Schaller et al., 2017). As it is, it is difficult to judge whether the data were processed in a proper way.

SPECIFIC COMMENTS

page 7, line 24-27 This part is unclear. Do you mean extreme outliers in the 20 Hz data or in the 1-min flux data? It is difficult to understand why the outliers could explain the difference in the amount of events observed with the two towers. Please clarify and rephrase.

Sect. 3.3 I would like to see an analysis using Richardson number (Ri), since Ri is typically used to indicate dynamic stability of the flow. Moreover, if Ri exceeds so called critical Richardson number (Ric) then the turbulence is strongly dampened or even almost completely wiped out (e.g. Grachev et al., 2013). Ric is typically said to be around 0.25, although this is debated (Galperin et al., 2007). It would be interesting to see how Ri is affected by these events you identified and if Ri>Ric always before the events. The analysis you did on stability parameter (z/L) is somewhat similar, however I would prefer Ri since you cannot determine a turbulence cutoff with z/L the same way as with Ri. I suggest you use gradient Richardson number for the analysis, however in case you are missing the needed vertical gradients, then use flux Richardson number.

p. 8, l. 20-25 Why the analysis with relative humidity? I would guess that it is not relevant for the topics at hand.

p. 8, l. 29-32 Referring to my comment before, did you do coordinate rotation? It should be always done, regardless of how flat the terrain is since anemometers are always at least slightly tilted. If coordinate rotation is not done, then w data is compromised by horizontal wind speed fluctuations.

p. 9, l. 6-8 It is difficult to understand why there would be unstable stratification during night. Could it be because during these events EC is not working properly and hence you have erroneous heat fluxes and therefore also erroneous values for z/L? Did you have also negative vertical gradients in air temperature (decrease with height) during these periods?

p. 9 l. 16 This title should be modified. Based on the evidence shown it is not possible to say that there was advection of CH4 to the study domain. In order to make such a statement you should have measured also CH4 concentration horizontal gradients.

Sect. 3.4 The event that is analyzed in this section was already analyzed in the companion paper (Schaller et al., 2017). For instance Fig. 4 here is partly the same as Fig. 5 in Schaller et al. (2017) and also the text is quite similar. It would be better to concentrate on some other event in this study, now this analysis is a bit redundant.

p. 10 l. 12-15 You analyse here a period that lasts for two hours, right? Can you then extent the maximum wavelet period above 120 min? If you can, then how accurate the results are at these very low frequencies, given that your time series does not cover even one whole wavelet when the wavelet period is above 120 min? On a related note, shouldn't you also take into account the cone of influence (those regions of the wavelet spectrum that are significantly affected by the edges; see Torrence and Compo, 1998) in your cross-scalograms and in the corresponding analysis? If you used two hour long time series in this analysis, then wavelet periods above 120 min are definitely within the cone of influence and hence unreliable.

p. 11 l. 4-5 This sentence should be modified. One cannot claim that EC fluxes were systematically overestimated since you do not have an absolutely correct reference. For instance damping of the signal within the cone of influence (Torrence and Compo, 1998) might decrease the wavelet based fluxes. This could partly explain the observed difference.

p. 11 l. 31 Please replace "by an eddy-covariance system" with "with these wavelet

algorithms".

Sect. 3.52 & 3.5.3 & 3.5.4 Difference between these three categories is difficult to see, especially the description of 3.5.3 and 3.5.4 looks similar. Try to emphasize more the differences in meteorological forcings between these event categories. As it reads now, combining the events with different mesoscale flow patterns seems rather subjective.

p. 12 l. 12-13 As you probably know, ebullition is often hypothesized to be connected with falling (Tokida et al., 2007), but sometimes also increasing atmospheric pressure (Chen and Slater, 2015). Could this be somehow connected to this daytime event?

p. 13 l. 20-21 Onset of turbulent mixing in the morning has been shown to cause CH4 flux peaks also in other studies (e.g. Peltola et al., 2015). Did these events that you identified to be connected with the onset of turbulent flow take place in the morning?

p. 14 l. 8 How did you define which events were influenced by advection? These periods discussed here are most likely non-stationary and would have been filtered out from standard EC data.

p. 14 l. 11-18 This is a good point and it would have been nice to see this idea used in the prior analysis.

p. 14 l. 24 How did you determine this 15 min limit for identifying events that are affected by advection?

Sect. 4.1.1 I would add here text about CH4 concentration profiles since large part of this manuscript discusses flushing of previously stored CH4 below the EC level. With detailed concentration profile you could measure this.

p. 15 l. 1-2 Why the analysis on cluster events was not possible?

Figure 2 Mean w is around 0.15 m/s, which is quite high value. Did you do coordinate rotation? You should definitely do it. Another thing: you could add here the Richardson number, like I suggest above.

Figures 2 & 3 These two figures are complicated and should be explained better. For instance how did you define "Unstable", "Stable" and "Neutral"? Where the stability is shown? How can you have EC data in the bottom plot with different quality classes at the same time?

Figures 2, 3 & 4 You most likely have change in flux sign at some certain color in the cross-scalograms (e.g. negative fluxes at blue colors and positive at red colors). Please highlight the zero flux lines in the cross-scalograms with e.g. white contour lines. Also, is the color scale the same in both subplots? If not, then please try to use one color scale per figure. Add also the cone of influence (Torrence and Compo, 1998) to all subplots.

TECHNICAL CORRECTIONS

p. 4, l. 12 You defined the abbreviation EC here, but you defined it already on page 2 line 19. Use the abbreviation everywhere in the text after you define it. Also, you use both "eddy covariance" and "eddy-covariance", replace both with EC.

p. 7, l. 14 and other places Please give dates in a consistent manner and try to follow the journal recommendations.

REFERENCES

Chen, X., and L. Slater (2015), Gas bubble transport and emissions for shallow peat from a northern peatland: The role of pressure changes and peat structure, Water Resources Research, 51(1), 151-168.

Galperin, B., et al. (2007), On the critical Richardson number in stably stratified turbulence, Atmospheric Science Letters, 8(3), 65-69.

Grachev, A. A., et al. (2013), The Critical Richardson Number and Limits of Applicability of Local Similarity Theory in the Stable Boundary Layer, Boundary-Layer Meteorology, 147(1), 51-82.

Peltola, O., et al. (2015), Studying the spatial variability of methane flux with five eddy covariance towers of varying height, Agricultural and Forest Meteorology, 214–215, 456-472.

Schaller, C., et al. (2017), Flux calculation of short turbulent events – comparison of three methods, Atmos. Meas. Tech., 10(3), 869-880.

Tokida, T., et al. (2007), Falling atmospheric pressure as a trigger for methane ebullition from peatland, Global Biogeochemical Cycles, 21(2), GB2003.

Torrence, C., and G. P. Compo (1998), A Practical Guide to Wavelet Analysis, B Am Meteorol Soc, 79(1), 61-78.

Wilczak, J. M., et al. (2001), Sonic Anemometer Tilt Correction Algorithms, Boundary-Layer Meteorology, 99(1), 127-150.

---

## Author Comment (AC1) · 30 Nov 2018

**Answer to N. Pirk**

*The comments of the reviewer are in black, our reply is coloured blue.*

The manuscript presents land-atmosphere methane fluxes from a permafrost-underlain wetland in NE Siberia, with a special focus on short-term fluctuations caused by non-turbulent conditions. The analysis uses wavelet transforms to calculate fluxes with higher time resolution than conventional eddy covariance calculations, also in non-turbulent conditions when the ground surface is (or has been) decoupled from the sensor level. Based on the wavelet flux time series, high-flux events are identified and classified according to their temporal structure. Some specific events are described in detail to distinguish the active mesoscale processes. The work should be interesting for many in the eddy covariance community and is relevant for the scope of ACP. The language and figures are of good quality. I therefore recommend the publication of this manuscript after minor revision considering my comments below.

We thank Norbert Pirk for his constructive comments. According to his remarks we revised our manuscript as described in the following reply.

1. Generally speaking, it is important to have studies that present alternatives to the conventional eddy covariance flux calculations. After reading this manuscript, however, I am left with the impression that wavelet flux estimations give arbitrary flux results. You used two different wavelets, the Mexican Hat and the Morlet, and get flux differences by about 30% on average (cf. Table 1). Other wavelets might have given even larger differences. I think this makes it difficult for the wider EC community to apply the wavelet analysis approach. While I understand that you cannot resolve this arbitrariness in your manuscript, I think you should be clearer about the potential and the shortcomings of wavelet flux calculations.

The reason for these differences in the calculated flux of the event shown in the case study in section 3.4 including Table 1 is directly connected to the mathematical behaviour of the mother wavelet. The Morlet wavelet allows a very good resolution in the frequency domain while its localization in the time domain is not that good (e.g. Collineau and Brunet, 1993). In our manuscript we used an averaging time of 30 minutes, which is widely used for eddy covariance processing. Usually this is not problematic in times of well-developed turbulence, because then the longest period contributing significantly to the flux should be 30 minutes (e.g. Charuchittipan et al., 2014; Foken et al., 2006). As already shown by Schaller et al., 2017, in times of well-developed turbulence, the results of both Mexican hat and Morlet wavelet are comparable and their deviation is within the typical error range of eddy covariance.

In the current manuscript we analyze fluxes also during times with only little or nearly no turbulence. Table 1 as well as Figure 4 (for tower 2) show the differences of the averaged flux magnitude over 30 minutes as mentioned by Norbert Pirk for the discussed event in section 3.4. The two upper panels show the wavelet cross scalograms for both wavelets. Exemplarily discussed for tower 2 and the Morlet wavelet, the area of high energy is sharp in frequency domain but shows a smearing effect in time domain from Aug 2, 23:00 to Aug 3, 0:45. The Mexican hat wavelet, on the other hand, resolves the event exactly in time domain (Aug 2, 23:59 – Aug 3, 0:07) while the resolution in frequency domain is slightly – but negligible – worse, compared to Morlet. These differences in resolution also explain the differences in the averaged flux over 30 minutes from 0:00 to 0:30. Both the Morlet wavelet flux (147 nmol $m^{-2}$ $s^{-1}$) and the eddy covariance flux (179 nmol $m^{-2}$ $s^{-1}$) underestimate the flux compared to Mexican hat (213 nmol $m^{-2}$ $s^{-1}$).

To obtain similar results at least for the two wavelets, it is necessary to extend the averaging period, i.e., the averaging period should be longer than the smallest contributing frequency. For the eddy covariance method it won't be possible in this case, because the steady-state assumption is not fulfilled.

To sum up: as long as we keep the fixed averaging period of 30 minutes, the Morlet wavelet is not able to resolve the event completely while the Mexican hat wavelet delivers exact and authoritative results. We agree to Norbert Pirk that the shortcomings of Morlet compared to Mexican hat wavelet were not described sufficiently in our manuscript. The findings and scalograms of the Morlet wavelet can be used as some kind of a complex measure for stationarity / steady-state-conditions, which extends the stationarity test after Foken and Wichura (1996) by an ogive test – but this might lead to misunderstandings; we also did not mention that clear enough in our manuscript.

As the aim of our paper is on the characterisation of single short-term events and not on the flux over a longer time, **we decided to remove the results of the Morlet wavelet**, because 1) they might confuse the reader and 2) they don't add significant information to the findings and results. Instead we will add a paragraph in section 2 about the choice of a proper mother wavelet (here: Mexican hat).

2. You imply on several occasions that the high-flux events you identified are related to methane emissions from the ecosystem. For example, when you relate event occurrences to soil temperatures in the abstract (page 1, lines 9ff, "We demonstrate..."). Also many other parts of your manuscript are written as if this study is about bio-physical processes, and not just about a different flux calculation method. At the same time, I think you are aware that your events are probably not ecosystem emissions, but merely a venting of previously accumulated methane. The fluxes you present, e.g. in Figure 1, indicate fast shifts between methane emission and uptake, which are unlikely to have anything to do with the ecosystem dynamics. There is a tendency throughout your manuscript to smear out the distinction between the ecosystem methane exchange and the flux you calculate at sensor level.

In fact, in our manuscript we also give a short information on the event seasonality (section 3.2), i.e. the number of occurrence of events during the time. It is known that increasing soil temperature changes the microbial decomposition rates of organic matter (e.g., Anderson, 1992; Valentine et al., 1994) and we could show that there is indeed a positive connection between the number of events and the soil temperature. Of course, the occurrence of meso-scale processes does not depend on soil temperature or other processes in the soil or the ecosystem, but it is an indirect measure for the activity of methanogenesis in soil that leads to a release of $CH_4$. So, e.g., during the observed maximum mean of the soil temperature in the first half of August, also the amount of methane released per time reached its maximum. E.g., during stable stratification in early August the biggest amount of CH4 gets accumulated per time close to the ground and later suddenly be released, in comparison to other months. **We will revise our manuscript so that this connection is clearly understandable.**

We agree that it is important to distinguish between methane emissions from the footprint of the EC system and emissions from around that were transported to the tower. Especially in the case of horizontal advection being the driver, the methane being vented to the EC system might originate partly from outside of the footprint. For regular flux calculations, especially for long-term balances and the discussion of biosphere-atmosphere interactions, this might definitely be a big issue. On the other hand, our investigations of advection in this study aim to show the impact of this meso-scale

trigger on the flux in a one minute resolution using wavelet analysis. **We will add some sentences to emphasize that the behaviour of the high resolution flux during the discussed advection event is not due to ecosystem dynamics but due to the venting of the accumulated and advected methane.**

3. On a related note, you mention that conventional EC processing would give biased budgets due to events of non-turbulent mixing, even if filtering and gap-filling is applied. However, you don't perform the comparison to a commonly-used filtering and gap-filling routine to make such a statement. If you want to say anything about such a possible bias, your analysis needs to show this.

Our manuscript explicitly investigates single events, that would usually be removed by gap-filling algorithms, but does not target long-term results. Such an extension of our manuscript by long-term studies including error analysis would be beyond the scope of this article. Nonetheless, we definitely agree that a comparison of wavelet fluxes and EC fluxes as well as EC gap filling approaches over a longer period should be performed to quantify its effect-term balances. This would be an interesting additional paper for future, which is already in preparation in our working group. **We will add this information to the revised manuscript; we will also revise the mentioned note, so that it is clear to the reader that we did not analyse this and that there will be a future paper analyzing the impact on long-term results in detail.**

4. You attribute parts of the high-flux events to methane that entered your footprint by horizontal advection. But since you have no direct measurements of horizontal methane advection, this attribution remains speculative in this study and should be phrased accordingly.

We agree, that it is much more difficult to detect advection without direct gradient profile measurements. Please see in general also our answer to your specific comment 7 concerning this point. Specifically to your question on advection events, Figure 6 in the manuscript supports this finding, because it directly shows the approaching fog bank. The occurrence and arrival of this fog bank at the towers exactly in time with the observed event in the data clearly indicates advection. Also in literature such kind of fog events are classified directly as advection fog (e.g., Stull, 1988).

5. I think your dataset of methane EC measurements from NE Siberia is impressive and extremely valuable, but in this manuscript you don't use this potential very much: you say and conclude little about this ecosystem or the methane dynamics of it. I understand that you want to focus on wavelet analysis and short-term events, but you could probably have done this for much easier field studies (something like CO2 exchange above a central European farmland). You don't even mention your field site location in the abstract. I think there is room for improvement to integrate and connect your findings to methane flux studies from permafrost wetlands.

Yes, we generally agree, that there are many easier ways to obtain datasets that contain influences of meso-scale processes like advection or weather fronts passing the site, e.g., in Germany instead of going to a very remote and difficultly accessible site in the Russian Far East. However, the original intention of our study was the identification of ebullition events using high frequency data from EC towers. Ebullition is a quite important $CH_4$ emission pathway in permafrost wetlands and therefore currently a big subject of research in the community. As you know from our manuscript, we could not encounter any signs of ebullition in the data. Instead we found wavelet analysis being capable to resolve periods of the dataset where the steady-state assumption was not fulfilled. While this

manuscript investigates specific events in detail, a follow-up paper will compare the wavelet results with common gap-filling routines, yielding information on its influence to the long-term balances.

Luckily, our manuscript is also not the only study that originates from this indeed extremly valuable dataset. So, based on that dataset, there are already published studies on the impact of a persistently lowered water table on $CO_2$ and $CH_4$ emissions (Kittler et al., 2016, 2017) as well as on the energy fluxes (Göckede et al., 2017). Additionally, the dataset was also already useful to compare results of a year-round $CH_4$ simulation model on $CH_4$ emissions with real data (Castro-Morales et al., 2018).

**We will emphasize the information on our follow-up paper and also add the field site location to the abstract.**

Specific comments:

6. Page 1, line 5 You mention that ebullition events last for only a few minutes, but I think the timescale of ebullition depends on the spatial scale. On a small spatial scale, maybe comprising a single bubble only, an ebullition event would probably only take seconds to be mixed into the ambient atmosphere.

Yes, the timescale of ebullition depends on the spatial scale. **We will change this sentence.**

7. Page 1, lines 12ff. "By investigating..." This sentence is unclear. You say you identified mesoscale processes as the dominating processes. But for what? You mean as the trigger for high-flux events? But then this is quite a stretch given your rather descriptive analysis of the mesoscale conditions.

Yes, we generally agree that it is difficult to link the found high flux events to the identified mesoscale processes. Your question is similar to general comment 3 by Anonymous Referee #2, which we both answer as follows:

Based on our meteorological measurements, we conclude that the identified mesoscale processes triggered the observed high-flux events. We agree that it is somehow "a stretch" or "speculative", to identify these mesoscale phenomena without measurements of spatial (vertical / horizontal) profiles. Usually, periods consisting of phenomena like the found mesoscale processes, are replaced by gap filling algorithms during the standard eddy covariance processing. Long-term measurements of the atmospheric boundary layer, including devices like SODAR-RASS or LIDAR as well as arrays of other vertical / horizontal gradient measurements, could fill this gap. Long-term measurements are necessary here, because these phenomena are not detected very often and any statistical analysis is nearly impossible then.

Nonetheless we think, that it will be worth to publish observations of such relatively rare phenomena if it is possible to observe them. The detection, identification and differentiation of such mesoscale phenomena requires considerable experience. Such experience is fortunately available in our working group, considering, e.g., the publications by Foken et al. (2012) or Serafimovich et al. (2018), where surface flux measurements and boundary-layer measurements were available. Of course, we carefully discussed the identified processes within our working group, to be as sure as possible in our statements regarding the available data. For the identification of these mesoscale processes we used the data from both towers (distance: 600 m). Here it is important to know that the identified phenomena are always visible at both tower 1 and 2, which supports our findings.  **This was not mentioned clearly in the manuscript, so we will emphasize this information.**

**To sum up, the classification of the different mesoscale processes is still not completely satisfying, but also not fully speculative. We will carefully revise the manuscript, so that it is clear for the reader why the identification is not just speculative.**

8. Page 1, lines 15f. "It is a reliable..." Please elaborate and clarify this statement. How exactly can I evaluate the flux quality using wavelets? And did you show that this works reliably?

Your question is quite similar to general comment 2) by Anonymous Referee #2. Thus, we answer both as follows: using a mother wavelet with an exact resolution in time domain like the Mexican hat wavelet yields the exact flux by integrating over a short time interval, e.g., 1 min. In this case also all parts in the frequency domain that contribute to the flux are included and considered (Percival and Walden, 2000). Summing up these 1 min fluxes to the typical EC averaging interval of 30 min, the results of both EC and wavelet method must be equal as long as the lowest contributing periods < 30 min. That is the case during steady-state conditions (Foken and Wichura, 1996; Foken et al., 2004, 2012) and Schaller et al. (2017) could prove that comparing both wavelet analysis and EC.

In the case of non steady-state conditions with contributing periods > 30 min, the EC quality control tests should flag those cases to be excluded (Foken et al., 2012). Additionally, in those cases also the ogive test (Desjardins et al., 1989; Foken et al., 1995; Oncley et al., 1990) yields contributions to the flux for periods > 30 min. Besides that, the Mexican hat wavelet will yield nonetheless in every case correct and trustworthy fluxes, also for periods > 30 min, if the integration interval in the period domain is chosen big enough (Percival and Walden, 2000; Torrence and Compo, 1998).

**We will add this information to the revised manuscript and modify the mentioned sentence.**

9. Page 4, line 22 What do you mean by "exact" fluxes?

The term "exact flux" in this context means that the result obtained by wavelet analysis offers an excellent resolution of the flux in time and frequency domain, depending on the chosen wavelet. **We will revise this sentence.**

10. Page 4, line 22 You mention the 1-min resolution of the flux results. But what is the limit for the time resolution of wavelet flux calculations? Why can you not resolve 1 sec, for example?

Please see the first paragraph of our answer to your comment 9. The maximum possible time resolution is restricted by the length of the timeseries and the cone of influence. Concerning the minimum resolution, we followed already available studies, because the exact calculation of the wavelet coefficients requires a certain number of values. A study on the lower limit of the resolution in time would be a specific mathematical study.

11. Page 4, line 25 Is the Morlet wavelet you used a real or complex function? I'm asking because I think in the PyWavelets Python package, the "Morlet" wavelet is real-valued, which might be unexpected. At the same time, a real-valued wavelet might have the advantage that you can show the flux direction (uptake/release) in your cross-scalograms (cf. Figure 3).

In our calculations we use the complex valued Morlet wavelet, but nonetheless the cross-wavelet scalogram only shows the real part of the complex number, which is also used by our flux calculations. As we will remove the Morlet wavelet investigations (see our answer to your general comment 1) from the revised manuscript, changes in our manuscript concerning this comment are not necessary.

12. Page 6, line 16 Here you mention that some event-minutes needed to be manually added. Later, in the last sentence of section 3.1, you describe the MAD test as a robust estimator. I wouldn't expect a robust test to need manual intervention.

Yes, the MAD test in general is a robust estimator for outliers / extreme values, but for some of the detected events, the method did not mark all non-extreme parts of the coherent event completely. In those cases we have made an additional visual inspection. **We will change the sentence to clarify that.**

13. Page 7, line 24 You explain the large difference between the two towers by the percentage of outliers. But what is the explanation for this difference in outliers? Could the real explanation be that tower 1's footprint was artificially drained?

In this section of our manuscript we refer to the statistics of detected events, i.e., the number of detected event-minutes. The statistics is based on the result of wavelet analysis, i.e., the 1-min Mexican hat wavelet flux.
At tower 1 7.7 % of all data were out of the range from $Q_1$-1.5($Q_3$-$Q_1$) to $Q_3$+1.5($Q_3$-$Q_1$), where $Q_1$ denotes the 25%- quantile and $Q_3$ denotes the 75%- quantile. At tower 2 5.4 % of the data are out of this range, i.e., 2.3 % less "outliers" than at tower 1.
As the MAD test is a robust estimator, it is quite resilient against such outliers. In consequence, due to the statistical properties of the data, the number of outliers (event minutes) detected by the MAD test is greater at tower 1. **We will modify the text in lines 24 – 27 to clarify that.**

Seen from ecology, the artifical drainage also has an impact on the measured flux, which leads to a reduction of the $CH_4$ emissions (Kittler et al., 2017). So the median flux at tower 1 (drained, 0.17 nmol mol$^{-1}$ m s$^{-1}$) was significantly smaller than at tower 2 (undrained, 0.60 nmol mol$^{-1}$ m s$^{-1}$). On the other hand, as written in our manuscript, the percentage of values exceeding the interquartile range by at least 1.5 times was 2.3 % (tower 1: 7.7 %, tower 2: 5.4 %). There are some possible explanations, but is difficult to prove them: 1) during stable or neutral stratification, advective processes transport methane from outside the drainage ditch into the footprint, which leads to a substantial increase of the $CH_4$ concentration. When flushing these accumulated $CH_4$ from the ground to the sensors, greater fluxes are caused, relatively to tower 2. 2) The drainage works quite good and generally lowers the water table up to 0.3 m in summer (Kittler et al., 2016; Kwon et al., 2017, 2016), but there are still some smaller patches within the drainage ditch and the footprint, where the water table was near the ground level. There the rate of methanogenesis might be greater than in other parts of the footprint, where it decreased. In times, where the footprint covers these wet patches, the flux might be greater in comparison to other wind directions. **In our manuscript, we did not include these ideas, because we could not prove them, so they are somehow speculative.**

14. Page 7, line 28 Here you describe the event seasonality, and I think it would be nice to see a plot of the event-percentages over time. You can show the three classes of events as separate lines, and the two towers as separate subplots. Maybe you can even add another subplot with the friction velocity.

As mentioned above, this paper exclusively focuses on the capability of our wavelet-based flux processing tool to detect and characterize events. Long-term statistics on event occurrence, including their net implication for the calculation of flux budgets, will be referred to the follow up paper mentioned before. We agree with the reviewer that it is somewhat disappointing to exclude the ecological implications of these event fluxes entirely within the context of this manuscript; however, we already know that those statistics cannot be explained with a short additional paragraph, so their proper interpretation would clearly exceed the appropriate length of this manuscript.

15. Page 8, line 16 How did you quantify or identify a trend?

In the context of p. 8, l. 16, "trend" just means that the values monotonically decrease, i.e. there is always are smaller value following the current value. Maybe, the word "trend" misleads the reader here, who might think that there is some kind of statistical trend modelling behind. **We will change this sentence and remove the word "trend" here.**

16. Page 9, line 34 I think it would be worthwhile to check if there is a relation be-
tween the length of the calm period preceding an event and the event's total emission.
This test could give you a needed insight to separate local emissions from horizontal
advection.

We had a look into this while developing the results for the original manuscript, but didn't find any conclusive correlation that would be worth to include into the paper.

17. Page 10, lines 14f If an extension of the upper period limit changes the flux so
much, does this mean there is no co-spectral gap? How does this problem look like
during well-mixed, stationary conditions? Have you looked at the ordinary co-spectrum
and ogive?

The Morlet cross-scalogram, which allows a high resolution in the frequency domain, does not show any signs of a cospectral gap (Fig. 5 of the original manuscript, bottom panel).
During well-mixed, stationary conditions, there are no significant differences observable between the flux for an upper period limit of each 30 min and 190 min, there is a co-spectral gap. Also ogive analysis supports the finding, that there are no flux contributions for periods > 30 min.
It should be noted that in most cases the error of the EC method and the additional flux contribution after calculating the ogives doesn't show significant differences (Charuchittipan et al., 2014).

18. Page 11, lines 4f Wouldn't the regular EC processing filter out and gap-fill this
period? If so, it doesn't seem right to say "regular EC data processing yielded biased
results". And have you checked that the momentum flux is downwards for all these
events you discuss here?

Yes, usually those times should not pass the tests on steady-state and/or turbulent conditions, so those times were filtered out. **We will change this sentence to clarify that.**
Under low wind velocities it is a difficult problem to identify if the momentum flux is upward or downward. The reason are flow distortion effects, as studied e.g. by Li et al. (2013).

19. Page 11, lines 19ff This whole paragraph hits the nail on the head. You should
focus on this finding in your abstract, instead of ebullition, which you probably didn't
observe.

**We will revise the abstract, so that ebullition does not play a dominant role any more and change the focus more to non-turbulent mixing.**

20. Page 14, line 20 Isn't it more the time since decoupling that determines how much methane can have accumulated, rather than the time since the last event?

**Yes, the time since decoupling should be the most important parameter, together with the time since the last event. We will add this to the manuscript.**

21. Page 15, lines 5f Methane budgets with the ecosystem as a reference should not include such high-flux events, because the ecosystem did not emit these large amounts of methane in this period. So I fail to see that filtering and gap-fill these periods would lead to a systematic underestimation of net emission, as you state here.

Yes, we definitely agree if you refer to advection as source for the measured event. The paragraph including p. 15, l. 5f starts on p. 14, l. 33 saying "In the absence of advection...". **To clarify for the reader that also the statement in p. 15, l. 5f is made under that assumption, we include "...provided that the event was not caused by advection..." into the sentence.** With advection being absent, the methane being vented during an event must have originated from the site itself, so these fluxes need to be included into the flux budget. Assuming rather constant local emissions, this would imply the tower 'sees' low-biased fluxes for a while before the event occurs, then the 'missing' methane is vented out at the onset of turbulence. Accordingly, filtering out the event would mean missing an important part of the long-term flux budget, even though not all the methane was produced at the time of venting.

22. Page 16, line 1 Why was this classification not possible here?

Your question is quite similar to the specific comment on p. 16 l 1-2 by Anonymous Referee #2. We answer both as follows: We decided to postpone this analysis to a follow-up study, because the complete, detailed investigation would be beyond the scope of this manuscript. It is quite difficult and due to the lack of additional boundary layer measurements maybe even impossible to get reliable evidence on the exact meso-scale processes triggering these events. Another major reason for our decision to exclude these events is the scope of the manuscript, which focuses on events that occur at short time scales, i.e., last only for minutes or some tens of minutes. **We agree, that these reasons were not stated clearly yet, and we will modify section 4.1.2 to make clear why the analysis might be possible, but is beyond our scope.**

23. Page 16, line 7 I'm not sure EC really "failed to resolve the events correctly". It is not designed to resolve them in the first place.

The EC method failed to resolve the events correctly, because it is not able to do that from design. **We will modify this sentence to clarify that.**

24. Page 16, line 16 How did you rule out sudden sources from the soil?

All of the observed events were triggered by meso-scale processes, mostly under stable or neutral stratification. Here the emitted methane was accumulated near the ground over longer times before the event started. During the occurrence of the meso-scale process, the accumulated methane was flushed up to the inlet of the measuring system tube causing the event flux. A differentiation whether the gas was accumulated over a longer time (very likely) or suddenly released directly from

the soil, is difficult. If it is a sudden release directly from soil, it is quite unlikely that it mostly happens during stable or neutral stratification as in our study.

25. Page 24, Figure 2 Your w-measurements seem to have a mean value of about 0.1 m/s, so this is data before the tilt correction? But your wavelet cross-scalograms use w after the tilt correction, right?

Anonymous Reviewer #2 also addressed this topic in his general comment #4. So we answer both your and his question as follows: Due to a sloped terrain, a non-exact alignment of the sonic anemometer or flow distortion effects, the streamlines of the wind might be tilted. In this case, usually a coordinate rotation is conducted to move the coordinate system into the streamlines and to solve this problem. For our study, we carefully inspected the measured wind vectors during times with well-developed turbulence and good flow conditions, i.e., sufficiently high wind velocities. We could not detect any disturbance of the streamlines due to the terrain or other influences, i.e., we are sure that the assumption of a negligible mean vertical wind component ($<w> = 0$) is valid. Due to that finding, the complete study is based on w-data without tilt correction.
Especially due to the fact that the found phenomena were connected to a distinct vertical wind component, double rotation might lead to irregularly high rotation angles (this was investigated in an master thesis by Matthias Mauder in 2002 for the EBEX-2000 experiment, flat terrain). Instead, planar fitting (Wilczak et al., 2001) is a proper choice in such situations, since here the rotation angles are based on a long term averaging period, so that short time periods, where $<w> \neq 0$, does not affect them. In our study we still did not apply planar fitting, because 1) it is not a long term study to make a careful analysis of the optimal interval for planar-fit rotation (Siebicke et al., 2012) and 2) at our towers fortunately the assumption of $<w> = 0$ was proven as valid for well-developed turbulence. **We will add this information to the revised manuscript.**

26. Page 25, Figure 3 The cross-scalograms don't seem to show a co-spectral peak, or an intensity decrease at the lowest and highest frequencies. Is this expected? Are these coefficients pre-multiplied by the frequency? Maybe a legend would help to read these plots. And did you define ITC and RNcov anywhere?

A co-spectral peak is not expected / visible in the cross-scalograms of this case study due to existent flux contributing periods > 30 min, which were shown in Fig. 5, where the upper period limit was set to 184 min.
The coefficients shown in all figures are not pre-multiplied by the frequency and show directly the wavelet coefficients. **We will change the colours of the plot, to make it possible to see directly the sign, i.e., the direction of flux.**
ITC and RNCov describe the integral turbulence characteristics and the steady state parameter, respectively (Foken et al., 2012). Both are very common and very well known in the EC community, so we did not define them in our manuscript.
**We will remove the panel for ITC in both Figures 3 and 4**, because ITC < 30 % was never reached in that time.

Technical corrections:
Thanks for your correction remarks, we will consider them in the revised manuscript.

Page 4, line 28 Missing full stop.
Page 8, line 22 You probably mean +0.67 % min-1
Page 10, line 5 Please add units to the fluxes given in parentheses

References

Anderson, J.M., 1992. Responses of Soils to Climate Change, in: Begon, M., Fitter, A.H., Macfadyen, A. (Eds.), The Ecological Consequences of Global Climate Change, Advances in Ecological Research. Academic Press, pp. 163–210. https://doi.org/10.1016/S0065-2504(08)60136-1

Castro-Morales, K., Kleinen, T., Kaiser, S., Zaehle, S., Kittler, F., Kwon, M.J., Beer, C., Göckede, M., 2018. Year-round simulated methane emissions from a permafrost ecosystem in Northeast Siberia. Biogeosciences 15, 2691–2722. https://doi.org/10.5194/bg-15-2691-2018

Charuchittipan, D., Babel, W., Mauder, M., Leps, J.-P., Foken, T., 2014. Extension of the Averaging Time in Eddy-Covariance Measurements and Its Effect on the Energy Balance Closure. Boundary-Layer Meteorology 152, 303–327. https://doi.org/10.1007/s10546-014-9922-6

Collineau, S., Brunet, Y., 1993. Detection of turbulent coherent motions in a forest canopy part I: Wavelet analysis. Boundary-Layer Meteorology 65, 357–379. https://doi.org/10.1007/BF00707033

Desjardins, R.L., Macpherson, J.I., Schuepp, P.H., Karanja, F., 1989. An Evaluation of Aircraft Flux Measurements of Co2, Water-Vapor and Sensible Heat. Boundary-Layer Meteorology 47, 55–69.

Foken, T., Dlugi, R., Kramm, G., 1995. On the determination of dry deposition and emission of gaseous compounds at the biosphere-atmosphere interface. Meteorol. Z. 91–118.

Foken, T., Göckede, M., Mauder, M., Mahrt, L., Amiro, B.D., Munger, J.W., 2004. Post-field data quality control, in: Lee, X., Massman, W., Law, B. (Eds.), Handbook of Micrometeorology: A Guide for Surface Flux Measurement and Analysis. Kluwer Academic Publishers, Dordrecht, pp. 181–208.

Foken, T., Leuning, R., Oncley, S.R., Mauder, M., Aubinet, M., 2012. Corrections and Data Quality Control, in: Eddy Covariance: A Practical Guide to Measurement and Data Analysis, Springer Atmospheric Sciences. Springer, Dordrecht, pp. 85–131.

Foken, T., Wichura, B., 1996. Tools for quality assessment of surface-based flux measurements. Agricultural and Forest Meteorology 78, 83–105. https://doi.org/10.1016/S1352-2310(96)00056-8

Foken, T., Wimmer, F., Mauder, M., Thomas, C., Liebethal, C., 2006. Some aspects of the energy balance closure problem. Atmos. Chem. Phys. 8.

Göckede, M., Kittler, F., Kwon, M.J., Burjack, I., Heimann, M., Kolle, O., Zimov, N., Zimov, S., 2017. Shifted energy fluxes, increased Bowen ratios, and reduced thaw depths linked with drainage-induced changes in permafrost ecosystem structure. The Cryosphere 11, 2975–2996. https://doi.org/10.5194/tc-11-2975-2017

Kittler, F., Burjack, I., Corradi, C.A.R., Heimann, M., Kolle, O., Merbold, L., Zimov, N., Zimov, S., Göckede, M., 2016. Impacts of a decadal drainage disturbance on surface–atmosphere fluxes of carbon dioxide in a permafrost ecosystem. Biogeosciences 13, 5315–5332. https://doi.org/10.5194/bg-13-5315-2016

Kittler, F., Heimann, M., Kolle, O., Zimov, N., Zimov, S., Göckede, M., 2017. Long-Term Drainage Reduces $CO_2$ Uptake and $CH_4$ Emissions in a Siberian Permafrost Ecosystem: Drainage impact on Arctic carbon cycle. Global Biogeochemical Cycles 31, 1704–1717. https://doi.org/10.1002/2017GB005774

Kwon, M.J., Beulig, F., Ilie, I., Wildner, M., Küsel, K., Merbold, L., Mahecha, M.D., Zimov, N., Zimov, S.A., Heimann, M., Schuur, E.A.G., Kostka, J.E., Kolle, O., Hilke, I., Göckede, M., 2017. Plants, microorganisms, and soil temperatures contribute to a decrease in methane fluxes on a drained Arctic floodplain. Global Change Biology 23, 2396–2412. https://doi.org/10.1111/gcb.13558

Kwon, M.J., Heimann, M., Kolle, O., Luus, K.A., Schuur, E.A.G., Zimov, N., Zimov, S.A., Göckede, M., 2016. Long-term drainage reduces CO2 uptake and

increases CO$_2$ emission on a Siberian floodplain due to shifts in vegetation community and soil thermal characteristics. Biogeosciences 13, 4219–4235. https://doi.org/10.5194/bg-13-4219-2016

Li, M., Babel, W., Tanaka, K., Foken, T., 2013. Note on the application of planar-fit rotation for non-omnidirectional sonic anemometers. Atmospheric Measurement Techniques 6, 221–229. https://doi.org/10.5194/amt-6-221-2013

Oncley, S.P., Businger, J.A., Itsweire, E.C., Friehe, C.A., Larue, J.C., Chang, S.S., 1990. Surface layer profiles and turbulence measurements over uniform land under near-neutral conditions, in: 9th Symp on Boundary Layer and Turbulence. Amer. Meteor. Soc., Roskilde, Denmark, pp. 237–240.

Percival, D.B., Walden, A.T., 2000. Wavelet methods for time series analysis. Cambridge University Press, Cambridge.

Schaller, C., Göckede, M., Foken, T., 2017. Flux calculation of short turbulent events -- comparison of three methods. Atmospheric Measurement Techniques 10, 869–880. https://doi.org/10.5194/amt-10-869-2017

Serafimovich, A., Metzger, S., Hartmann, J., Kohnert, K., Zona, D., Sachs, T., 2018. Upscaling surface energy fluxes over the North Slope of Alaska using airborne eddy-covariance measurements and environmental response functions. Atmospheric Chemistry and Physics 18, 10007–10023. https://doi.org/10.5194/acp-18-10007-2018

Stull, R.B., 1988. An Introduction to Boundary Layer Meteorology. Kluwer Academic Publishers, Dordrecht, Boston, London.

Torrence, C., Compo, G.P., 1998. A Practical Guide to Wavelet Analysis. Bulletin of the American Meteorological Society 79, 61–78. https://doi.org/10.1175/1520-0477(1998)079<0061:APGTWA>2.0.CO;2

Valentine, D.W., Holland, E.A., Schimel, D.S., 1994. Ecosystem and physiological controls over methane production in northern wetlands. Journal of Geophysical Research: Atmospheres 99, 1563–1571. https://doi.org/10.1029/93JD00391

---

## Author Comment (AC2) · 30 Nov 2018

The comment was uploaded in the form of a supplement:
https://www.atmos-chem-phys-discuss.net/acp-2018-277/acp-2018-277-AC2-supplement.pdf

[Figure]

**Answer to Anonymous Referee #2**

*The comments of the reviewer are in black, our reply is coloured blue.*

This manuscript presents an analysis of methane (CH4) eddy covariance (EC) data measured above a wetland in NE Siberia. The manuscript focuses on CH4 fluxes during night time in non-turbulent and low-mixing conditions when the EC measurement level is decoupled from the surface. Wavelet methods developed in a companion paper are used to estimate fluxes with 1 min time resolution over one summer and this high frequency flux time series is used to identify and classify high CH4 flux events during the analyzed period. These events are then speculated to be linked with atmospheric mesoscale circulation taking place in these nocturnal low-mixing conditions. However, large part of the abstract, introduction and some other sections are discussing ebullition and other non-related topics, whereas results and conclusions are all about nocturnal low-mixing conditions. The authors should modify the beginning of the manuscript so that it matches with the end, so that the text forms one coherent entity. There are also other shortcomings in the text and description of data processing. Please see below.

As it stands the manuscript is interesting and shows promise but requires major revisions (see below) before publication. Once revised, it should be of interest also for the wider community working with micrometeorological flux measurements and hence the study is within the scope of ACP. Besides the shortcomings mentioned above, the presentation quality is good, although some figures need adjustment. I recommend the publication of this manuscript after major revision based on the comments below.

We thank Anonymous Referee #2 for his constructive comments. According to his remarks we revised our manuscript as described in the following reply.

GENERAL COMMENTS
1) Please modify the abstract and introduction so that they match with the results. In my opinion these sections should be largely rewritten since now they are quite disconnected from the rest of the manuscript. The results are about gas fluxes under nocturnal low-mixing conditions and the abstract and introduction should be written about this topic, not about arctic wetland CH4 emission dynamics. As you know, these problems related to low-mixing conditions are universal, not only related to arctic wetlands.

We agree that there is some kind of disconnection between the specific process of ebullition, which is presented in abstract and introduction, and the results of our manuscript. Nonetheless we think that it is important to consider that the scientific discussion on methane emissions in Arctic permafrost wetlands mentions ebullition as an important pathway. Thus the main reason of our data analysis was to find signs of ebullition using the wavelet approach – in our case studies, we detected other reasons for all found events, but no signs of ebullition. It seems that ebullition, occurring as heterogeneous single events on the spatial scale of the EC footprint of our towers, is not detectable. We think, that this finding might be also important for the scientific community. **We will rewrite parts of abstract and introduction as requested, so that it will be clear that ebullition is not the main topic of the manuscript, but we decided not to remove our remarks on ebullition completely due to its importance in Arctic permafrost wetlands.**

2) The wavelet method is presented in the manuscript as more accurate than EC and

**Fig. 1.**

**Supplement:**

**Answer to Anonymous Referee #2**

*The comments of the reviewer are in black, our reply is coloured blue.*

This manuscript presents an analysis of methane (CH4) eddy covariance (EC) data measured above a wetland in NE Siberia. The manuscript focuses on CH4 fluxes during night time in non-turbulent and low-mixing conditions when the EC measurement level is decoupled from the surface. Wavelet methods developed in a companion paper are used to estimate fluxes with 1 min time resolution over one summer and this high frequency flux time series is used to identify and classify high CH4 flux events during the analyzed period. These events are then speculated to be linked with atmospheric mesoscale circulation taking place in these nocturnal low-mixing conditions. However, large part of the abstract, introduction and some other sections are discussing ebullition and other non-related topics, whereas results and conclusions are all about nocturnal low-mixing conditions. The authors should modify the beginning of the manuscript so that it matches with the end, so that the text forms one coherent entity. There are also other shortcomings in the text and description of data processing. Please see below.

As it stands the manuscript is interesting and shows promise but requires major revisions (see below) before publication. Once revised, it should be of interest also for the wider community working with micrometeorological flux measurements and hence the study is within the scope of ACP. Besides the shortcomings mentioned above, the presentation quality is good, although some figures need adjustment. I recommend the publication of this manuscript after major revision based on the comments below.

*We thank Anonymous Referee #2 for his constructive comments. According to his remarks we revised our manuscript as described in the following reply.*

GENERAL COMMENTS
1) Please modify the abstract and introduction so that they match with the results. In my opinion these sections should be largely rewritten since now they are quite disconnected from the rest of the manuscript. The results are about gas fluxes under nocturnal low-mixing conditions and the abstract and introduction should be written about this topic, not about arctic wetland CH4 emission dynamics. As you know, these problems related to low-mixing conditions are universal, not only related to arctic wetlands.

*We agree that there is some kind of disconnection between the specific process of ebullition, which is presented in abstract and introduction, and the results of our manuscript. Nonetheless we think that it is important to consider that the scientific discussion on methane emissions in Arctic permafrost wetlands mentions ebullition as an important pathway. Thus the main reason of our data analysis was to find signs of ebullition using the wavelet approach – in our case studies, we detected other reasons for all found events, but no signs of ebullition. It seems that ebullition, occurring as heterogeneous single events on the spatial scale of the EC footprint of our towers, is not detectable. We think, that this finding might be also important for the scientific community.*
***We will rewrite parts of abstract and introduction as requested, so that it will be clear that ebullition is not the main topic of the manuscript, but we decided not to remove our remarks on ebullition completely due to its importance in Arctic permafrost wetlands.***

2) The wavelet method is presented in the manuscript as more accurate than EC and

reliable reference for the EC fluxes. This is a strong statement, which should be supported by convincing evidence. In order to make that kind of statement you should show that when the data is processed using standard EC methodologies spurious data points are left in the flux time series, yet with the wavelet methods these problematic periods are handled better. I suspect that most of the low-mixing conditions would have been filtered out by quality screening and friction velocity filtering the data. Hence CH4 budgets derived using standard EC processing are likely not affected by these spurious fluxes during low-mixing. Please show a comparison of fluxes (e.g. monthly CH4 budgets) derived with standard EC processing (including quality screening and friction velocity filtering) and fluxes calculated with your wavelet methods to support your statement. Alternatively, you should phrase the text differently so that it is clear for the reader that it is not possible to say that the wavelet method is more accurate or reliable than standard EC.

Your question is quite similar to specific comment 8 by Norbert Pirk. Thus, we answer both as follows: using a mother wavelet with an exact resolution in time domain like the Mexican hat wavelet yields the exact flux by integrating over a short time interval, e.g., 1 min. In this case also all parts in the frequency domain that contribute to the flux are included and considered (Percival and Walden, 2000). Summing up these 1 min fluxes to the typical EC averaging interval of 30 min, the results of both EC and wavelet method must be equal as long as the lowest contributing periods < 30 min. That is the case during steady-state conditions (Foken and Wichura, 1996; Foken et al., 2004, 2012) and Schaller et al. (2017) could prove that comparing both wavelet analysis and EC.

In the case of non steady-state conditions with contributing periods > 30 min, the EC quality control tests should flag those cases to be excluded (Foken et al., 2012). Additionally, in those cases also the ogive test (Desjardins et al., 1989; Foken et al., 1995; Oncley et al., 1990) yields contributions to the flux for periods > 30 min. Besides that, the Mexican hat wavelet will yield nonetheless in every case correct and trustworthy fluxes, also for periods > 30 min, if the integration interval in the period domain is chosen big enough (Percival and Walden, 2000; Torrence and Compo, 1998).

**We will add this information to the revised manuscript.**

As you correctly mention, the event cases in our study would be excluded and/or replaced most probably during the gap filling analysis due to poor quality flags or the friction velocity u* being too low. Usually, gap filling algorithms determine the regression by binning the data into classes and calculating the median. This procedure leads to an exclusion of "outliers" containing very big or very small fluxes – and especially these usually rejected "outliers" which might contain distinctive, significant fluxes are targeted in our manuscript. There are a few phenomena that could explain such processes, like ebullition, free or wet convection, low level jets, breaking gravity waves or advection during calm wind, which have partly also been addressed in previous single studies.

Our manuscript explicitly investigates single events that would usually be removed by gap-filling algorithms, but does not target long-term results. A long-term study using EC on our dataset was already published by Kittler et al. (2017). An extension of our manuscript by long-term studies on the wavelet flux including error analysis would be beyond the scope of this article. Nonetheless, we definitely agree that a comparison of wavelet fluxes and EC fluxes as well as EC gap filling approaches over a longer period should be performed to quantify its effect-term balances. This would be an interesting additional future paper, which is already in preparation in our working group. **We will add this information to the revised manuscript.**

3) In many occasions the observed high flux events are linked with mesoscale motions (gravity waves, low-level jets etc.), yet the connections are quite speculative. This is understandable since these mesoscale motions are difficult to quantify with just one flux tower and the authors also clearly state this in the discussion section of the manuscript. However in contrast to the discussion, the abstract and the conclusions are written in such way that the connections are obvious based on the data. Please rephrase the text so that it is clear for the reader that the role of mesoscale motions is quite speculative and additional instrumentation would be needed for a proper identification of these flow patterns.

Yes, we generally agree that it is difficult to link the found high flux events to the identified mesoscale processes. Your question is similar to specific comment 7 by reviewer Norbert Pirk, which we both answer as follows:

Based on our meteorological measurements, we conclude that the identified mesoscale processes triggered the observed high-flux events. We agree that it is somehow "a stretch" or "speculative", to identify these mesoscale phenomena without measurements of spatial (vertical / horizontal) profiles. Usually, periods consisting of phenomena like the found mesoscale processes, are replaced by gap filling algorithms during the standard eddy covariance processing. Long-term measurements of the atmospheric boundary layer, including devices like SODAR-RASS or LIDAR as well as arrays of other vertical / horizontal gradient measurements, could fill this gap. Long-term measurements are necessary here, because these phenomena are not detected very often and any statistical analysis is nearly impossible then.

Nonetheless we think, that it will be worth to publish observations of such relatively rare phenomena if it is possible to observe them. The detection, identification and differentiation of such mesoscale phenomena require considerable experience. Such experience is fortunately available in our working group, considering, e.g., the publications by Foken et al., (2012) or Serafimovich et al. (2018), where surface flux measurements and boundary-layer measurements were available. Of course, we carefully discussed the identified processes within our working group, to be as sure as possible in our statements regarding the available data. For the identification of these mesoscale processes we used the data from both towers (distance: 600 m). Here it is important to know that the identified phenomena are always visible at both tower 1 and 2, which supports our findings. **This was mentioned in the manuscript only in the first sentence of section 3.3, so we will add this finding also to the conclusion to highlight it.**

To sum up, the classification of the different mesoscale processes is not completely satisfying, but also not fully speculative. **We will carefully revise the manuscript, so that it is clear for the reader why the identification is not just speculative.**

4) More details about data processing are needed. Did you do coordinate rotation and how did you do it? The regular 2D-coordinate rotation (align u with mean wind and nullify mean w for each 30 min period) does not necessarily work well during low-turbulence since mean w is not necessarily zero when mesoscale motions are at play. Hence I hope that you used planar fitting (Wilczak et al., 2001) and defined the plane used in the coordinate rotation using high quality data. On the other hand if you did not do any coordinate rotation (like in Schaller et al., 2017) then the fluxes might be seriously compromised, since sonic anemometers are always slightly tilted no matter how carefully they are aligned with the surface below. Also did you correct for the time lag between gas analyzer data and sonic data? Lag time determination is always difficult for periods with low and intermittent turbulence. Therefore, please

provide additional details about EC processing. Related to the wavelet analysis, how did you take into account the time series edges and their effect on the results? Did you zero-pad the data and then estimate the cone of influence (Torrence and Compo, 1998)? This is important especially for the low frequencies. Please add details, since they are missing also from the companion paper (Schaller et al., 2017). As it is, it is difficult to judge whether the data were processed in a proper way.

Thanks for addressing these important points, we agree that in the current state of the manuscript it is difficult to be sure that the data (pre-) processing was done in a completely correct way, so we will add these information as follows.

Coordinate rotation – Norbert Pirk also addressed this topic in his specific comment #25. So we answer both your and his question as follows: Due to a sloped terrain, a non-exact alignment of the sonic anemometer or flow distortion effects, the streamlines of the wind might be tilted. In this case, usually a coordinate rotation is conducted to move the coordinate system into the streamlines and to solve this problem. For our study, we carefully inspected the measured wind vectors during times with well-developed turbulence and good flow conditions, i.e., sufficiently high wind velocities. We could not detect any disturbance of the streamlines due to the terrain or other influences, i.e., we are sure that the assumption of a negligible mean vertical wind component ($<w> = 0$) is valid. Due to that finding, the complete study is based on w-data without tilt correction.
Especially due to the fact that the found phenomena were connected to a distinct vertical wind component, double rotation might lead to irregularly high rotation angles (this was investigated in an master thesis by Matthias Mauder in 2002 for the EBEX-2000 experiment, flat terrain). Instead, planar fitting (Wilczak et al., 2001) is a proper choice in such situations, since here the rotation angles are based on a long term averaging period, so that short time periods, where $<w> \neq 0$, does not affect them. In our study we still did not apply planar fitting, because 1) it is not a long term study to make a careful analysis of the optimal interval for planar-fit rotation (Siebicke et al., 2012) and 2) at our towers fortunately the assumption of $<w> = 0$ was proven as valid for well-developed turbulence. **We will add this information to the revised manuscript.**

Time lag correction – the time lag correction between the sonic anemometer and the gas analyser was conducted by maximisation of the covariances by cross correlation for every 30-minutes-interval. Because of the conditions with low turbulence the time lag may be different for each time series. No constant time lag was applied. **We will add this information to the revised manuscript.**

Edge effects in wavelet analysis – the filtering of erroneous results of the wavelet calculations at the beginning and end of the cross scalogram is of crucial importance to obtain results of high quality without border effects (Torrence and Compo, 1998). Our wavelet analysis was conducted on the whole available dataset from 1[st] June to 15[th] September 2014 using a windowed approach. For the Mexican hat wavelet, the lag between two subsequent window calculation start points was set to 6 hours while the length of each calculated window was 12 hours, i.e., it overlaps the window start/end points. The data was zero-padded and the cone of influence estimated. At the end, the calculated windows were cut at the borders and merged, so that border effects in the final result only occur at the beginning and end of the whole time series. These border times were excluded from further analysis. **We will add this information to the revised manuscript.**

SPECIFIC COMMENTS

page 7, line 24-27 This part is unclear. Do you mean extreme outliers in the 20 Hz data or in the 1-min flux data? It is difficult to understand why the outliers could explain the difference in the amount of events observed with the two towers. Please clarify and rephrase.

In this section of our manuscript we refer to the statistics of detected events, i.e., the number of detected event-minutes. The statistics is based on the result of wavelet analysis, i.e., the 1-min Mexican hat wavelet flux.
At tower 1 7.7 % of all data were out of the range from $Q_1$-1.5($Q_3$-$Q_1$) to $Q_3$+1.5($Q_3$-$Q_1$), where $Q_1$ denotes the 25%- quantile and $Q_3$ denotes the 75%- quantile. At tower 2 5.4 % of the data are out of this range, i.e., 2.3 % less "outliers" than at tower 1.
As the MAD test is a robust estimator, it is quite resilient against such outliers. In consequence, due to the statistical properties of the data, the number of outliers (event minutes) detected by the MAD test is greater at tower 1. **We will modify the text in lines 24 – 27 to clarify that.**

Sect. 3.3 I would like to see an analysis using Richardson number (Ri), since Ri is typically used to indicate dynamic stability of the flow. Moreover, if Ri exceeds so called critical Richardson number (Ric) then the turbulence is strongly dampened or even almost completely wiped out (e.g. Grachev et al., 2013). Ric is typically said to be around 0.25, although this is debated (Galperin et al., 2007). It would be interesting to see how Ri is affected by these events you identified and if Ri>Ric always before the events. The analysis you did on stability parameter (z/L) is somewhat similar, however I would prefer Ri since you cannot determine a turbulence cutoff with z/L the same way as with Ri. I suggest you use gradient Richardson number for the analysis, however in case you are missing the needed vertical gradients, then use flux Richardson number.

The flux Richardson number (Rf) can be converted exactly into z/L (e.g., Arya, 2001; Foken, 2017); the gradient Richardson number should not be used under stable conditions because of possible decoupling between the both measurement heights. It can be assumed that the critical Richardson number is approximately equal to z/L=−1. The calculation for Rf and z/L is extremely affected by errors because for stable conditions u or u* are near zero, and small errors can generate large effects on the stability parameter. For the accuracy under such conditions see also Högström (1996). Additionally, in our studies we use z/L only as a general classifier for the stability, i.e., the exact numerical values should not have an important impact on our findings.

p. 8, l. 20-25 Why the analysis with relative humidity? I would guess that it is not relevant for the topics at hand.

Especially ecologists use the relative humidity (combined with air temperature) quite often for studies on atmosphere-biosphere exchange. We think that the information on humidity provides meaningful information that should remain in the manuscript.

p. 8, l. 29-32 Referring to my comment before, did you do coordinate rotation? It should be always done, regardless of how flat the terrain is since anemometers are always at least slightly tilted. If coordinate rotation is not done, then w data is compromised by horizontal wind speed fluctuations.

Please see our answer to your general comment 4).

p. 9, l. 6-8 It is difficult to understand why there would be unstable stratification during night. Could it be because during these events EC is not working properly and hence you have erroneous heat fluxes and therefore also erroneous values for z/L? Did you have also negative vertical gradients in air temperature (decrease with height) during these periods?

We agree that usually during times with short-wave incoming radiation ≈ 0 unstable stratification is quite unlikely to happen. In the high-latitude Arctic zone (68.78° N), depending on the date, there are just about a few hours between sunset and sunrise. In our study, we defined "night" by the fixed time span 21:00 to 9:00. **We will add this information to the revised manuscript.**

Additionally, we revisited all events that occurred during our fixed nighttime span 21:00 to 9:00. Unstable stratification was observed only in times where also short-wave incoming radiation > 20 Wm$^{-2}$ was measured. In such cases it is not unlikely to observe at least slightly unstable situations. **We will add this information to the revised manuscript.**

p. 9 l. 16 This title should be modified. Based on the evidence shown it is not possible to say that there was advection of CH4 to the study domain. In order to make such a statement you should have measured also CH4 concentration horizontal gradients.

Please see in general our answer to your general comment 3, where we show why our statements on the mesoscale processes are still not completely satisfying, but definitely not fully speculative. Due to the fact that there are only rare studies on such kind of mesoscale events in the Arctic, we think that it is worth to publish them.
Specifically to your question on section 3.4 labeled "Nighttime advection": We agree, that it is much more difficult to detect advection without direct gradient profile measurements. On the other hand, Figure 6 in the manuscript supports this finding and gives evidence, because it directly shows the approaching fog bank. The occurrence and arrival of this fog bank at the towers exactly in time with the observed event in the data clearly indicates advection. Also in literature such kinds of fog events are classified directly as advection fog (e.g., Stull, 1988).

Sect. 3.4 The event that is analyzed in this section was already analyzed in the companion paper (Schaller et al., 2017). For instance Fig. 4 here is partly the same as Fig. 5 in Schaller et al. (2017) and also the text is quite similar. It would be better to concentrate on some other event in this study, now this analysis is a bit redundant.

We agree that there is a bit of redundancy between this manuscript and Schaller et al. (2017), due to the fact that the event discussed intensively in our manuscript was already shown partly there. But, the focus between the two publications is completely different: Schaller et al. (2017) show the methodology behind the wavelet flux calculation and validate the method against EC for times of well-developed turbulence and stationary conditions. In section 3.2.2 of that publication we gave a very short insight to the capability of wavelet analysis to resolve also fluxes in times, where the steady-state condition is not fulfilled. In that manuscript, we "promised" the reader that there will be a follow-up paper, which investigates such kind of events as well as this specific event in detail. We think that we should keep this "promise" we gave in our companion paper, so we decided not to change our focus to another event.
**In the revised paper we will clarify that this example was analyzed in** Schaller et al. (2017) **with respect to the method, while this article investigates the underlying process.**

p. 10 l. 12-15 You analyse here a period that lasts for two hours, right? Can you then extent the maximum wavelet period above 120 min? If you can, then how accurate the results are at these very low frequencies, given that your time series does not cover even one whole wavelet when the wavelet period is above 120 min? On a related note, shouldn't you also take into account the cone of influence (those regions of the wavelet spectrum that are significantly affected by the edges; see Torrence and Compo, 1998) in your cross-scalograms and in the corresponding analysis? If you used two hour long time series in this analysis, then wavelet periods above 120 min are definitely within the cone of influence and hence unreliable.

Please see our remarks on the edge effects on your general comment 4). If there are only two hours of data, a wavelet analysis on this dataset will not yield to trustworthy data of that length due to parts of the results being within the cone of influence. None of the data we used or showed in our studies are influenced by edge / border effects, as already explained. **We will add this information to the revised manuscript.**

p. 11 l. 4-5 This sentence should be modified. One cannot claim that EC fluxes were systematically overestimated since you do not have an absolutely correct reference. For instance damping of the signal within the cone of influence (Torrence and Compo, 1998) might decrease the wavelet based fluxes. This could partly explain the observed difference.

Please see our answer on your general comment 2). Due to the mathematical properties of the wavelet analysis in general the steady-state assumption has not to be fulfilled, while eddy covariance results might be significantly biased on the same time. Additionally, the chosen mother wavelet (Mexican hat) allows a very good resolution in the time domain and thus, as there is no influence by edge effects in this period, there is no damping of the signal caused by the cone of influence. **It is not possible to set up an "absolutely correct reference" under field conditions. To consider that, we remove the word "strongly" in the revised manuscript.**

p. 11 l. 31 Please replace "by an eddy-covariance system" with "with these wavelet algorithms".

Because the footprint does not depend on the data analysis we believe that "EC system" is already a more general name than "EC method" or "wavelet method". **To be sure that the reader will not misunderstand that, we changed the sentence to "by this EC setup with a sensor height ≥ 4.9 m above ground".**

Sect. 3.52 & 3.5.3 & 3.5.4 Difference between these three categories is difficult to see, especially the description of 3.5.3 and 3.5.4 looks similar. Try to emphasize more the differences in meteorological forcings between these event categories. As it reads now, combining the events with different mesoscale flow patterns seems rather subjective.

Please see our answer to your general comment 3), which also addresses the differentiation between the meteorological forcings. Concerning the mentioned sections, we carefully revisited them: the examples 3.5.1, 3.5.3, 3.5.4 look similar, but the reason is different. 3.5.1 is due to the constant wind direction and the observed fog clearly an advection situation. 3.5.3 and 3.5.4 show both an increase of turbulence, but a change of the wind direction is typical for a low level jet and not for

braking gravity waves. Example 3.5.2 has all criteria for a passage of a frontal system, see e.g. pressure. After carefully revisiting the events, **we decided to keep the differentiation, also due to our answer to your general comment 3).**

p. 12 l. 12-13 As you probably know, ebullition is often hypothesized to be connected with falling (Tokida et al., 2007), but sometimes also increasing atmospheric pressure (Chen and Slater, 2015). Could this be somehow connected to this daytime event?

We thoroughly inspected the observations for that daytime event again. The course of the air pressure, which was decreasing before the event and increasing by 2 hPa/hour with and after the event clearly supports the assumption of a front passing by, which is also observable in all fluxes, not only the methane flux.

p. 13 l. 20-21 Onset of turbulent mixing in the morning has been shown to cause CH4 flux peaks also in other studies (e.g. Peltola et al., 2015). Did these events that you identified to be connected with the onset of turbulent flow take place in the morning?

All events that were determined to be caused by the onset of turbulent flow were observed during night time, but none of them in the morning. It should be noted here that due to high geographical latitudes there was never a complete sunset down to darkness and also the sunrise does not occur that rapidly as, e.g., in the middle latitudes like in Central Europe.

p. 14 l. 8 How did you define which events were influenced by advection? These periods discussed here are most likely non-stationary and would have been filtered out from standard EC data.

Please see our answer to your general comment 3). In the usual EC processing these periods would be filtered out, and subsequently treated by gap filling algorithms as they are in fact non-stationary and would fail the steady-state test.

p. 14 l. 11-18 This is a good point and it would have been nice to see this idea used in the prior analysis.

Such an analysis needs an extended experimental setup. We think that this analysis should be done using data from a long-term study including more towers, additional boundary layer measurements as well as arrays of other vertical / horizontal gradient measurements.

p. 14 l. 24 How did you determine this 15 min limit for identifying events that are affected by advection?

In fact, this limit is somehow a helpful "practical rule of thumb" for the events observed at our measuring site considering the available. On the other hand, we also see that is difficult or even impossible to specify an exact minimum event duration, where you can be sure that advection is the driver. A long-term study would be necessary to prove that, **therefore we will remove this sentence in our manuscript.**

Sect. 4.1.1 I would add here text about CH4 concentration profiles since large part of this manuscript discusses flushing of previously stored CH4 below the EC level. With detailed concentration profile you could measure this.

Yes, profile measurements would be definitely a good measure to support findings on meso-scale processes like in our study. In addition to our answer to your general comment 3), **we will also add this information to our manuscript.**

p. 15 l. 1-2 Why the analysis on cluster events was not possible?

We assume, you mean p. 16 l. 1-2. We decided to postpone this analysis to a follow-up study, because the complete, detailed investigation would be beyond the scope of this manuscript. It is quite difficult and due to the lack in additional boundary layer measurements maybe even impossible to get reliable evidence on the exact meso-scale processes triggering these events. Another major reason for our decision to exclude these events is the scope of the manuscript, which focuses on events that occur at short timescales, i.e., last only for minutes or some tens of minutes. **We agree, that this reasons were not stated clearly yet, so we will modify section 4.1.2, so that is clear, why the analysis might be possible but beyond the scope.**

Figure 2 Mean w is around 0.15 m/s, which is quite high value. Did you do coordinate rotation? You should definitely do it. Another thing: you could add here the Richardson number, like I suggest above.

Please see our answer on your general remark 4) and on your specific remark on section 3.3.

Figures 2 & 3 These two figures are complicated and should be explained better. For instance how did you define "Unstable", "Stable" and "Neutral"? Where the stability is shown? How can you have EC data in the bottom plot with different quality classes at the same time?

**We will improve the caption text of both figures, so that everything is well explained.** For the definition of the stability we followed the recommendations on error analysis by Foken and Wichura (1996) and Foken et al. (2004, 2012) to be consistent. These recommendations derive from the fact that here the universal functions are nearly equal to 1 (Foken and Skeib, 1983). Stability was labeled "unstable" for $zL^{-1} < -0.0625$, "stable" for $zL^{-1} > 0.0625$, "neutral" for $-0.0625 \leq zL^{-1} \leq 0.0625$. In the plot the stability is shown for every 30-minute-interval directly underneath the cross-wavelet scalogram of Mexican hat.

The quality class of the EC data is labeled by small rectangles in the bottom panel of Fig. 4f for each 30 minute interval, i.e., the timestep 0:00 – 0:30 in Fig. 3 is labeled as quality class 4 – 6. There are no different quality classes at the same time, but maybe here the Figure is misleading the reader. **Therefore, we will revise the bottom planel, so that is easier to determine the EC quality class.**

Figures 2, 3 & 4 You most likely have change in flux sign at some certain color in the cross-scalograms (e.g. negative fluxes at blue colors and positive at red colors). Please highlight the zero flux lines in the cross-scalograms with e.g. white contour

lines. Also, is the color scale the same in both subplots? If not, then please try to use one color scale per figure. Add also the cone of influence (Torrence and Compo, 1998) to all subplots.

Yes, the values in cross-scalogram are negative or positive valued, depending on the direction of flux. **We will change the colors in the revised manuscript, so that intensity and algebraic sign are easy to read off. Also the color scale for the subplots in Figures 2, 3 & 4 will be changed. There is no cone of influence (see your general comment 4), but we will note this in figure captions.**

TECHNICAL CORRECTIONS

p. 4, l. 12 You defined the abbreviation EC here, but you defined it already on page 2 line 19. Use the abbreviation everywhere in the text after you define it. Also, you use both "eddy covariance" and "eddy-covariance", replace both with EC.

We will change that.

p. 7, l. 14 and other places Please give dates in a consistent manner and try to follow the journal recommendations.

We will change that.

References

Arya, S.P., 2001. Introduction to Micrometeorology, 2nd ed. Academic Press, San Diego.

Desjardins, R.L., Macpherson, J.I., Schuepp, P.H., Karanja, F., 1989. An Evaluation of Aircraft Flux Measurements of Co2, Water-Vapor and Sensible Heat. Boundary-Layer Meteorology 47, 55–69.

Foken, T., 2017. Micrometeorology, 2nd ed. Springer, Berlin.

Foken, T., Dlugi, R., Kramm, G., 1995. On the determination of dry deposition and emission of gaseous compounds at the biosphere-atmosphere interface. Meteorol. Z. 91–118.

Foken, T., Göckede, M., Mauder, M., Mahrt, L., Amiro, B.D., Munger, J.W., 2004. Post-field data quality control, in: Lee, X., Massman, W., Law, B. (Eds.), Handbook of Micrometeorology: A Guide for Surface Flux Measurement and Analysis. Kluwer Academic Publishers, Dordrecht, pp. 181–208.

Foken, T., Leuning, R., Oncley, S.R., Mauder, M., Aubinet, M., 2012. Corrections and Data Quality Control, in: Eddy Covariance: A Practical Guide to Measurement and Data Analysis, Springer Atmospheric Sciences. Springer, Dordrecht, pp. 85–131.

Foken, T., Skeib, G., 1983. Profile measurements in the atmospheric near-surface layer and the use of suitable universal functions for the determination of the turbulent energy exchange. Boundary-Layer Meteorology 25, 55–62. https://doi.org/10.1007/BF00122097

Foken, T., Wichura, B., 1996. Tools for quality assessment of surface-based flux measurements. Agricultural and Forest Meteorology 78, 83–105. https://doi.org/10.1016/S1352-2310(96)00056-8

Högström, U., 1996. Review of some basic characteristics of the atmospheric surface layer. Boundary-Layer Meteorology 78, 215–246. https://doi.org/10.1007/BF00120937

Kittler, F., Heimann, M., Kolle, O., Zimov, N., Zimov, S., Göckede, M., 2017. Long-Term Drainage Reduces $CO_2$ Uptake and $CH_4$ Emissions in a Siberian Permafrost Ecosystem: Drainage

impact on Arctic carbon cycle. Global Biogeochemical Cycles 31, 1704–1717. https://doi.org/10.1002/2017GB005774

Oncley, S.P., Businger, J.A., Itsweire, E.C., Friehe, C.A., Larue, J.C., Chang, S.S., 1990. Surface layer profiles and turbulence measurements over uniform land under near-neutral conditions, in: 9th Symp on Boundary Layer and Turbulence. Amer. Meteor. Soc., Roskilde, Denmark, pp. 237–240.

Percival, D.B., Walden, A.T., 2000. Wavelet methods for time series analysis. Cambridge University Press, Cambridge.

Schaller, C., Göckede, M., Foken, T., 2017. Flux calculation of short turbulent events -- comparison of three methods. Atmospheric Measurement Techniques 10, 869–880. https://doi.org/10.5194/amt-10-869-2017

Serafimovich, A., Metzger, S., Hartmann, J., Kohnert, K., Zona, D., Sachs, T., 2018. Upscaling surface energy fluxes over the North Slope of Alaska using airborne eddy-covariance measurements and environmental response functions. Atmospheric Chemistry and Physics 18, 10007–10023. https://doi.org/10.5194/acp-18-10007-2018

Siebicke, L., Hunner, M., Foken, T., 2012. Aspects of CO2 advection measurements. Theoretical and Applied Climatology 109, 109–131. https://doi.org/10.1007/s00704-011-0552-3

Stull, R.B., 1988. An Introduction to Boundary Layer Meteorology. Kluwer Academic Publishers, Dordrecht, Boston, London.

Torrence, C., Compo, G.P., 1998. A Practical Guide to Wavelet Analysis. Bulletin of the American Meteorological Society 79, 61–78. https://doi.org/10.1175/1520-0477(1998)079<0061:APGTWA>2.0.CO;2

Wilczak, J.M., Oncley, S.P., Stage, S.A., 2001. Sonic anemometer tilt correction algorithms. Boundary-Layer Meteorology 99, 127–150. https://doi.org/10.1023/A:1018966204465

---

## Referee Report (RR1)

Second review of " Characterisation of short-term extreme methane fluxes related to non-turbulent mixing above an Arctic permafrost ecosystem" by Schaller et al. acp-2018-277

The manuscript has improved compared to the previous version and the authors have addressed most of the reviewer comments in a satisfactory manner. Now it is clearer throughout the text that the identified events are related to meteorology and not changes in surface methane ($CH_4$) emission and most of the other issues were taken care as well. Good job.

However, one possibly critical issue still remains. The authors did not do any coordinate rotation to their eddy covariance (EC) data, which might significantly impact the results. The authors claim in their response that under high wind velocities mean vertical wind component (<w>) was negligible meaning that their sonic coordinate frame was not tilted respect to the local flow stream lines. However, this claim is not backed by empirical evidence (e.g. figure). If coordinates are not rotated, not only the mean value of w, but also the turbulent fluctuations of w are compromised by horizontal wind speed, which in turn affect the estimated fluxes. In order to emphasise this point, consider Eq. (6.13) in Kaimal and Finnigan (1994):

$$w_3 = -u_2 \sin \varphi + w_2 \cos \varphi, \tag{1}$$

where $w_3$ is the vertical wind component in such coordinates that $\overline{w_3} = 0$ (overbar denotes temporal averaging), $u_2$ is the horizontal wind component in such direction that $\overline{u_2} = U$, i.e. the mean equals mean horizontal wind speed ($U$), $w_2$ is the unrotated vertical wind component (i.e. $w$ in the sonic coordinate frame) and

$$\varphi = \tan^{-1} \left( \frac{\overline{w_2}}{\overline{u_2}} \right). \tag{2}$$

Here $w_2$ equals the vertical wind speed data in the same coordinate frame as used in this study. Now if we use Reynolds decomposition (primes denote fluctuations around the mean), multiply the Eq. (1) with scalar fluctuations ($c'$), take the temporal average and reorganize the terms, we get

$$\overline{w_2'c'} = \overline{u_2'c'} \tan \varphi + \overline{w_3'c'} \frac{1}{\cos \varphi}. \tag{3}$$

Meaning that the horizontal wind speed fluctuations are directly affecting the fluxes if they are calculated utilizing $w_2$ as done in this study. Maybe even more importantly, it is unclear how the spectral decomposition of $\overline{w_3'c'}$ and $\overline{w_2'c'}$ differ from each other, meaning that the coordinate rotation might have different effect on turbulent fluxes at different frequencies. This would have an effect especially on the interpretation of results derived using the wavelet decomposition of the signal. Therefore, it is essential to show that the sonic was not indeed tilted, meaning that $\varphi$ is zero in all wind directions.

I suggest that the authors show a figure where $\overline{w_2}$ is plotted against $U$ in different wind direction bins. If there is no significant dependence between $\overline{w_2}$ and $U$ ($\varphi$ values at max around 2° or so), then no coordinate rotation is needed, yet if there is a dependence, then coordinate rotation is needed. In such case planar fit coordinate rotation method (Wilczak et al., 2001) should be implemented in the data processing chain and the results should be recalculated.

Based on Fig. 2 in the manuscript, $\varphi$ was around 6° if we assume that the mean w in the figure (0.15 m s$^{-1}$) is only related to the tilted coordinate frame. In my view, this would already warrant implementation of coordinate rotation to the processing chain.

MINOR REMARKS:

page 3, lines 9-10. This part should be modified since the EC method is doing exactly this: calculating $CH_4$ fluxes directly from high frequency EC measurements.

p. 3, l. 27 Long-term $CH_4$ budgets were not analysed in this study. You only discuss the possible effect that these events may have on the long-term budgets, but do not show any hard data. Hence, please reformulate this sentence.

p. 10, l. 13-26 and maybe elsewhere Please use only one format for time (not both 11:00pm and 23:00) and follow the journal recommendations.

REFERENCES

Kaimal, J. C. and Finnigan, J. J.: Atmospheric boundary layer flows : their structure and measurement, Oxford University Press, New York., 1994.

Wilczak, J. M., Oncley, S. P. and Stage, S. A.: Sonic Anemometer Tilt Correction Algorithms, Boundary-Layer Meteorol., 99(1), 127–150, doi:10.1023/A:1018966204465, 2001.

---

## Author Response (AR2)

**Answer to Anonymous Referee #2**

We thank Anonymous Referee #2 for the acceptance of most of our corrections from the first round of revisions. Since our comments to the problem of coordinate rotation were not accepted, we will explain our rationale in this document in more detail. As references to support our line of argumentation, in our previous reply we used mainly our own books and peer-reviewed manuscripts (e.g. Siebicke et al., 2012; Foken, 2017), supported by long-time experiences. In the extended version presented here, we instead use references of authors that are not members of our groups.

The reviewer proposed to apply double rotation (DR) for our measurements (Kaimal and Finnigan, 1994). This method was validated in the last 25 years by many authors and it is widely used in measurement programs in combination with the application of a gapfilling routine after $u^*$-threshold filtering. Regarding this coordinate rotation, a widely-cited recent monograph on the eddy-covariance method (Aubinet et al., 2012) states on page 77, lines 14-23 "However, drawbacks of the DR rotation procedure became apparent …. Limitations are … degradation of data quality (unrealistically large pitch angles in low wind speed conditions)". Now, low wind speed conditions dominate our measurements! We tried to explain this problem in our first answer on the reviewers, but obviously failed to clearly make our point here. We therefore hope that the given reference by Aubinet et al. – an update of an earlier textbook (Lee et al., 2004b) – convinces the reviewer that the double rotation cannot be applied on our measurements.

With the application of the DR excluded, there would still be the option to apply planar-fit (PF) coordinate rotation (Wilczak et al., 2001) to our dataset. Regarding this method, Aubinet et al. (2012) comment on page 78, lines 19-21: "It is recommended to reject low wind speed conditions (generally below 1 m s$^{-1}$) for the computation of the regression coefficients, thereby removing the problem of unrealistically large pitch angles." But the method can be applied for all wind velocities.

Because of the characteristics of the landscape surrounding our measurement site on the Kolyma River floodplain near Chersky, we followed another strategy. Since as a recent floodplain, the terrain is absolutely flat, and we took care that the anemometer is orientated exactly in the same plane as the surface, mass flow divergences can be ignored (see Aubinet et al. (2012), page 74, lines 7-20). We believe that the statement we have provided in our manuscript (p. 5, lines 2-4) clearly describes this setup: "There was no tilt in the alignment of the sonic anemometers at both towers and after a careful inspection of the raw data no disturbances of the streamlines due to the terrain or other influences could be found. This allows the assumption that <w> = 0 for well-developed turbulence". In other words, when <w> = 0 for well-developed turbulence, also the planar-fit method would result in no rotation angle, since low wind conditions would have been excluded for angle calculation.

Summarizing, double rotation cannot be applied under low wind conditions, and due to our undisturbed streamlines under well-developed turbulence there is no need to apply planar-fit rotation. Besides this fact, in general we question the relevance of this discussion about coordinate rotation effects on the findings presented within our manuscript. Even assuming a permanent misalignment of the wind sensor with tilt angles not exceeding 2°, the resulting errors for scalar fluxes would be lower than 5 % (see p. 75, lines 2-5, in Aubinet et al (2012), which again is based on material from p. 52-53 (Lee et al., 2004b). So even for a moderate misalignment of the sensor, errors would be far less than other errors of the eddy covariance method  (see e.g. Aubinet et al. (2012), p. 85-131 - sorry, this is again a self-citation). Since we can make the claim that our rotation angles are close to zero, we are confident that any error that may be associated with a potential <w> <> 0 would be very small, and thus would not alter the interpretation of our results.

There doesn't seem to be an argumentation by the reviewer that an update of the present state of art (Lee et al., 2004b; Aubinet et al., 2012; Foken, 2017; Rebmann et al., 2018) is available. Therefore we do not see the necessity to modify our paper. Still, we agree to include an additional sentence on p.5 line 5, in our manuscript, stating: "Small tilt errors have no significant influences on scalar fluxes (Lee et al., 2004a)".

Minor Remarks

p. 3 lines 9-10: we will include in this sentence "with a time resolution of about 1 minute".

p. 3 line 27: we agree with the reviewer that, based on the material presented in the current paper, long-term effects cannot be evaluated. We therefore shortened this sentence to "We present an analysis of whether peak $CH_4$ emission events at timescales on the order of minutes can be found in the results, and what their basic characteristics are."

p. 10, lines 13-26: we will correct this mismatch in timestamps throughout the manuscript.

[revised manuscript text omitted]